# TACTiS-2: Better, Faster, Simpler Attentional Copulas for Multivariate Time Series

**Arjun Ashok**[1 2 3], [*]**Étienne Marcotte**[1], [*]**Valentina Zantedeschi**[1],
[†]**Nicolas Chapados**[1 2], [†]**Alexandre Drouin**[1 2]
[1]ServiceNow Research [2]Mila-Quebec AI Institute [3]Université de Montréal
Montréal, Canada
`firstname.lastname@servicenow.com`

## Abstract

We introduce a new model for multivariate probabilistic time series prediction, designed to flexibly address a range of tasks including forecasting, interpolation, and their combinations. Building on copula theory, we propose a simplified objective for the recently-introduced *transformer-based attentional copulas* (TACTiS), wherein the number of distributional parameters now scales linearly with the number of variables instead of factorially. The new objective requires the introduction of a training curriculum, which goes hand-in-hand with necessary changes to the original architecture. We show that the resulting model has significantly better training dynamics and achieves state-of-the-art performance across diverse real-world forecasting tasks, while maintaining the flexibility of prior work, such as seamless handling of unaligned and unevenly-sampled time series. Code is made available at https://github.com/ServiceNow/TACTiS.

## 1 Introduction

Optimal decision-making involves reasoning about the evolution of quantities of interest over time, as well as the likelihood of various scenarios (Peterson, 2017). In its most general form, this problem amounts to estimating the joint distribution of a set of variables over multiple time steps, i.e., *multivariate probabilistic time series forecasting* (Gneiting & Katzfuss, 2014). Although individual aspects of this problem have been extensively studied by the statistical and machine learning communities (Hyndman et al., 2008; Box et al., 2015; Hyndman & Athanasopoulos, 2018), they have often been examined in isolation. Recently, an emerging stream of research has started to seek general-purpose models that can handle several stylized facts of real-world time series problems, namely (i) a large number of time series, (ii) arbitrarily complex data distributions, (iii) heterogeneous or irregular sampling frequencies (Shukla & Marlin, 2020), (iv) missing data (Fang & Wang, 2020), and (v) the availability of deterministic covariates for conditioning (e.g., holiday indicators), while being flexible enough to handle a variety of tasks, such as forecasting and interpolation (Drouin et al., 2022).

Classical forecasting methods (Hyndman et al., 2008; Box et al., 2015; Hyndman & Athanasopoulos, 2018) often make strong assumptions about the nature of the data distribution (e.g., parametric forms) and are thus limited in their handling of these desiderata. The advent of deep learning-based methods enabled significant progress on this front and led to models that excel at a wide range of time series prediction tasks (Torres et al., 2021; Lim & Zohren, 2021; Fang & Wang, 2020). Yet, most of these methods lack the flexibility required to meet the aforementioned requirements. Recently, some general-purpose models have been introduced (Tashiro et al., 2021; Drouin et al., 2022; Biloš et al., 2023; Alcaraz & Strodthoff, 2023), but most of them only address a subset of the desiderata. One notable exception is *Transformer-Attentional Copulas for Time Series* (TACTiS; Drouin et al., 2022), which addresses all of them, while achieving state-of-the-art predictive performance. TACTiS relies on a modular *copula-based* factorization of the predictive joint distribution (Sklar, 1959), where multivariate dependencies are modeled using *attentional copulas*. These consist of neural networks trained by solving a specialized objective that guarantees their convergence to mathematically valid copulas. While the approach of Drouin et al. (2022) is theoretically sound, it requires solving an optimization problem with a number of distributional parameters that grows factorially with the number of variables, leading to poor training dynamics and suboptimal predictions.

---

[*]Equal Contribution, [†]Equal Contribution

In this paper, we build on copula theory to propose a simplified training procedure for *attentional copulas*, solving a problem where the number of distributional parameters scales linearly, instead of factorially, with the number of variables. Our work results in a general-purpose approach to multivariate time series prediction that also meets all of the above desiderata, while having considerably better training dynamics, namely converging faster to better solutions (see Fig. 1).

**Contributions:**

- We show that, while nonparametric copulas require specialized learning procedures (Prop. 1), the permutation-based approach used in TACTiS (Drouin et al., 2022) is unnecessarily complex, and that valid copulas can be learned by solving a two-stage problem whose number of parameters scales linearly with the number of variables (Sec. 3.2);

- We build on these theoretical findings to propose TACTiS-2, an improved version of TACTiS with a revised architecture that is trained using a two-stage curriculum guaranteed to result in valid copulas (Sec. 4);

- We empirically show that our simplified training procedure leads to better training dynamics (e.g., faster convergence to better solutions) as well as state-of-the-art performance on a number of real-world forecasting tasks, while preserving the high flexibility of the TACTiS model (Sec. 5);

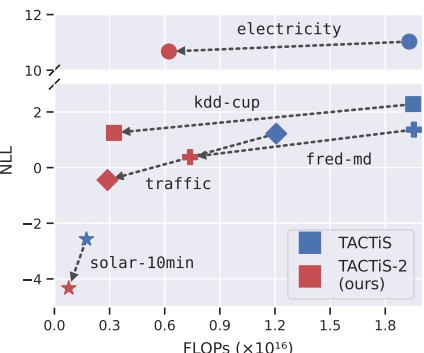

**Figure 1:** TACTiS-2 *outperforms* TACTiS *in (i) density estimation (lower validation negative log-likelihoods, NLL) and (ii) training compute (fewer floating point operations, FLOPs) in real-world forecasting tasks (see Sec. 5).*

## 2 PROBLEM SETTING

This work addresses the general problem of estimating the joint distribution of unobserved values at arbitrary time points in multivariate time series. This setting subsumes classical ones, such as forecasting, interpolation, and backcasting. Let $\mathbf{X}$ be a multivariate time series comprising of $n$, possibly related, univariate time series, denoted as $\mathbf{X} \overset{\text{def}}{=} \{\mathbf{X}_1, \ldots, \mathbf{X}_n\}$. Each $\mathbf{X}_i \overset{\text{def}}{=} [X_{i1}, \ldots, X_{i,\ell_i}]$ is a random vector representing $\ell_i$ observations of some real-valued process in time. We assume that, for any realization $\mathbf{x}_i \overset{\text{def}}{=} [x_{i1}, \ldots, x_{i,\ell_i}]$ of $\mathbf{X}_i$, each $x_{ij}$ is paired with (i) a timestamp, $t_{ij} \in \mathbb{R}$, with $t_{ij} < t_{i,j+1}$, marking its measurement time and (ii) a vector of non-stochastic covariates $\mathbf{c}_{ij} \in \mathbb{R}^p$ that represents arbitrary additional information available at each time step.

**Tasks:** We define our learning tasks with the help of a mask $m_{ij} \in \{0, 1\}$, which determines if any $X_{ij}$ should be considered as observed ($m_{ij} = 1$) or to be inferred ($m_{ij} = 0$). For example, a task that consists in forecasting the last $k$ time steps in $\mathbf{X}$ would be defined as $m_{ij} = 0$ for all $i$ and $j$, s.t. $\ell_i - k < j \leq \ell_i$ and $m_{ij} = 1$ otherwise. Similarly, a task that consists in interpolating the values of time steps $k$ to $p$ in $\mathbf{X}$ could be defined by setting all $m_{ij} = 0$ for all $i$ and $j$ s.t. $k \leq j \leq p$ and $m_{ij} = 1$ otherwise. Arbitrary, more complex, tasks can be defined using this approach.

**General Problem:** We consider the general problem of estimating the joint distribution of *missing values* (i.e., $m_{ij} = 0$), given the observed ones ($m_{ij} = 1$), the covariates, and the timestamps:

$$P\Big(\mathbf{X}^{(m)} \;\Big|\; \mathbf{X}^{(o)}, \mathbf{C}^{(m)}, \mathbf{C}^{(o)}, \mathbf{T}^{(m)}, \mathbf{T}^{(o)}\Big), \tag{1}$$

where $\mathbf{X}^{(m)} = [X_{11}, \ldots, X_{1,l_1}; \ldots; X_{n1}, \ldots, X_{n,l_n} \mid m_{ij} = 0]$ is a random vector containing the $d$ random variables $X_{ij}$ corresponding to all missing values, $\mathbf{X}^{(o)}$ is the same, but for observed values, and $\mathbf{C}^{(m)}, \mathbf{C}^{(o)}, \mathbf{T}^{(m)}, \mathbf{T}^{(o)}$ correspond to identical partitionings of the covariates and timestamps, respectively. In what follows, we estimate the joint probability density function (PDF) of Eq. (1) using a *copula-based density estimator* $g_\phi(x_1, \ldots, x_d)$, whose parameters $\phi$ are conditioned on $\mathbf{X}^{(o)}, \mathbf{C}^{(m)}, \mathbf{C}^{(o)}, \mathbf{T}^{(m)}, \mathbf{T}^{(o)}$.

### 2.1 COPULA-BASED DENSITY ESTIMATORS

A copula is a probabilistic object that allows capturing dependencies between $d$ random variables independently from their respective marginal distributions. More formally the joint cumulative distribution function

(CDF) for any random vector of $d$ variables $\mathbf{X} = [X_1, \ldots, X_d]$, can be written as:

$$P(X_1 \leq x_1, \ldots, X_d \leq x_d) = C\big(F_1(x_1), \ldots, F_d(x_d)\big), \tag{2}$$

where $F_i(x_i) \stackrel{\text{def}}{=} P(X_i \leq x_i)$ is the *univariate* marginal CDF of $X_i$ and $C : [0,1]^d \to [0,1]$, the *copula*, is the CDF of a *multivariate* distribution on the unit cube with *uniform* marginals (Sklar, 1959). Notably, if all $F_i$ are continuous, then this copula-based decomposition is unique.

The present work integrates into a stream of research seeking specific parametrizations of copula-based density estimators for multivariate time series (Salinas et al., 2019; Drouin et al., 2022), namely,

$$g_{\boldsymbol{\phi}}(x_1, \ldots, x_d) \stackrel{\text{def}}{=} c_{\phi_c}\Big(F_{\phi_1}(x_1), \ldots, F_{\phi_d}(x_d)\Big) \times f_{\phi_1}(x_1) \times \cdots \times f_{\phi_d}(x_d), \tag{3}$$

where $\boldsymbol{\phi} = \{\phi_1, \ldots, \phi_d; \phi_c\}$, with $\{\phi_i\}_{i=1}^d$ the parameters of the marginal distributions (with $F_{\phi_i}$ and $f_{\phi_i}$ the estimated CDF and PDF respectively) and $\phi_c$ the parameters of the copula density $c_{\phi_c}$. The choice of distributions for $F_{\phi_i}$ and $c_{\phi_c}$ is typically left to the practitioner. For example, one could take $F_{\phi_i}$ to be the CDF of a Gaussian distribution and $c_{\phi_c}$ to be a *Gaussian copula* (Nelsen, 2007). The parameters can then be estimated by minimizing the negative log-likelihood:

$$\underset{\boldsymbol{\phi}}{\arg\min} \; - \underset{\mathbf{x} \sim \mathbf{X}}{\mathbb{E}} \log g_{\boldsymbol{\phi}}(x_1, \ldots, x_d) \,. \tag{4}$$

## 3    LEARNING NONPARAMETRIC COPULAS

Oftentimes, to avoid making parametric assumptions, one may take the $c_{\phi_c}$ and $F_{\phi_i}$ components of the estimator to be highly flexible neural networks (Wiese et al., 2019; Janke et al., 2021; Drouin et al., 2022). While it is easy to constrain $c_{\phi_c}$ to have a valid domain and codomain, this does not imply that its marginal distributions will be uniform. Thus, as observed by Janke et al. (2021) and Drouin et al. (2022), a key challenge is ensuring that the distribution $c_{\phi_c}$ satisfies the mathematical definition of a copula (see Sec. 2.1). We strengthen this observation by proving a new theoretical result, showing that solving Problem (4) without any additional constraints can lead to infinitely many solutions where $c_{\phi_c}$ is not a valid copula:

**Proposition 1.** *(Invalid Solutions) Assuming that all random variables $X_1, \ldots, X_d$ have continuous marginal distributions and assuming infinite expressivity for $\{F_{\phi_i}\}_{i=1}^d$ and $c_{\phi_c}$, Problem (4) has infinitely many invalid solutions wherein $c_{\phi_c}$ is not the density function of a valid copula.*

*Proof.* The proof, detailed in App. B.1, shows that one can create infinitely many instantiations of $F_{\phi_i}$ and $c_{\phi_c}$ where $p(x_1, \ldots, x_d) = g_{\boldsymbol{\phi}}(x_1, \ldots, x_d)$, but the true marginals and the copula are entangled. $\qquad\square$

Hence, using neural networks to learn copula-based density estimators is non-trivial and requires more than a simple modular parametrization of the model.

### 3.1    PERMUTATION-INVARIANT COPULAS

Recently, Drouin et al. (2022) showed that valid nonparametric copulas can be learned using a permutation-based objective. Their approach considers a factorization of $c_{\phi_c}$ based on the chain rule of probability according to an arbitrary permutation of the variables $\boldsymbol{\pi} = [\pi_1, \ldots, \pi_d] \in \Pi$, where $\Pi$ is the set of all $d!$ permutations of $\{1, \ldots, d\}$. The resulting copula density, $c_{\phi_c^{\boldsymbol{\pi}}}$, can be written as:

$$c_{\boldsymbol{\phi}_c^{\boldsymbol{\pi}}}(u_1, \ldots, u_d) \stackrel{\text{def}}{=} c_{\phi_{c,1}^{\boldsymbol{\pi}}}(u_{\pi_1}) \times c_{\phi_{c,2}^{\boldsymbol{\pi}}}(u_{\pi_2} \mid u_{\pi_1}) \times \cdots \times c_{\phi_{c,d}^{\boldsymbol{\pi}}}\big(u_{\pi_d} \mid u_{\pi_1}, \ldots, u_{\pi_{d-1}}\big), \tag{5}$$

where $u_{\pi_k} = F_{\phi_{\pi_k}}(x_{\pi_k})$, with $F_{\phi_{\pi_k}}$ arbitrary marginal CDFs, and the $c_{\phi_{c,i}^{\boldsymbol{\pi}}}$ are arbitrary distributions (e.g., histograms) on the unit interval with parameters $\phi_{c,i}^{\boldsymbol{\pi}}$. There is, however, one important exception: the density of the first variable in the permutation $c_{\phi_{c,1}^{\boldsymbol{\pi}}}$ is always taken to be that of a *uniform distribution* $U_{[0,1]}$ and thus $c_{\phi_{c,1}^{\boldsymbol{\pi}}}(u_{\pi_1}) = 1$. This choice, combined with solving the following problem, guarantees that, at the minimum, all of the $c_{\phi_c^{\boldsymbol{\pi}}}$, irrespective of $\boldsymbol{\pi}$, are equivalent and correspond to valid copula densities:

$$\underset{\phi_1, \ldots, \phi_d, \phi_c^{\boldsymbol{\pi}}}{\arg\min} \; - \underset{\mathbf{x} \sim \mathbf{X}}{\mathbb{E}} \; \underset{\boldsymbol{\pi} \sim \Pi}{\mathbb{E}} \log c_{\phi_c^{\boldsymbol{\pi}}}\Big(F_{\phi_1}(x_1), \ldots, F_{\phi_d}(x_d)\Big) \times f_{\phi_1}(x_1) \times \cdots \times f_{\phi_d}(x_d), \tag{6}$$

where $\phi_c^{\boldsymbol{\pi}} \stackrel{\text{def}}{=} \{\phi_{c,1}^{\boldsymbol{\pi}}, \dots \phi_{c,d}^{\boldsymbol{\pi}}\}_{\boldsymbol{\pi} \in \Pi}$ are the parameters for each of the $d!$ factorizations of the copula density (see Drouin et al. (2022), Theorem 1).

**Limitations:** Obtaining a valid copula using this approach requires solving an optimization problem with $O(d!)$ parameters. This is extremely prohibitive for large $d$, which are common in multivariate time series prediction (e.g., $d = 8880$ for the common `electricity` benchmark; Marcotte et al. (2023)). The approach proposed in Drouin et al. (2022), consists of parametrizing a single neural network to output $\boldsymbol{\phi}^\Pi$ and using a Monte Carlo approximation of the expectation on $\Pi$. However, such an approach has several caveats, e.g., (i) the neural network must have sufficient capacity to produce $O(d!)$ distinct values and (ii) due to the sheer size of $\Pi$, only a minuscule fraction of all permutations can be observed in one training batch, resulting in slow convergence rates. This is supported by empirical observations in Sec. 5.

## 3.2 Two-Stage Copulas

In this work, we take a different approach to learning nonparametric copulas, one that does not rely on permutations, alleviating the aforementioned limitations. Our approach builds on the following two-stage optimization problem, whose properties have previously been studied in the context of parametric (Joe & Xu, 1996; Andersen, 2005; Joe, 2005) and semi-parametric estimators (Andersen, 2005), and where the number of parameters scales with $O(d)$ instead of $O(d!)$:

$$\underset{\phi_c}{\arg\min} \quad - \underset{\mathbf{x} \sim \mathbf{X}}{\mathbb{E}} \log c_{\phi_c}\left(F_{\phi_1^\star}(x_1), \dots, F_{\phi_d^\star}(x_d)\right) \tag{7}$$

$$s.t. \qquad (\phi_1^\star, \dots, \phi_d^\star) \in \underset{\phi_1, \dots, \phi_d}{\arg\min} - \underset{\mathbf{x} \sim \mathbf{X}}{\mathbb{E}} \log \prod_{i=1}^{d} f_{\phi_i}(x_i). \tag{8}$$

Hence, the optimization proceeds in two stages:

> **Stage 1:** Learn the marginal parameters, irrespective of multivariate dependencies (Eq. (8));

> **Stage 2:** Learn the copula parameters, given the optimal marginals (Eq. (7)).

Beyond obtaining an optimization problem that is considerably simpler, we show that any nonparametric copula $c_{\phi_c}$ learned using this approach is valid, i.e., that it satisfies the mathematical definition of a copula:

**Proposition 2.** *(Validity) Assuming that all random variables $X_1, \dots, X_d$ have continuous marginal distributions and assuming infinite expressivity for $\{F_{\phi_i}\}_{i=1}^{d}$ and $c_{\phi_c}$, solving Problem (7) yields a solution to Problem (4) where $c_{\phi_c}$ is a valid copula.*

*Proof.* The proof builds on results from Sklar (1959) and is provided in App. B.2. □

## 4 The TACTiS-2 Model

Building on Sec. 3.2, we propose TACTiS-2, a model for multivariate probabilistic time series prediction that inherits the flexibility of the Drouin et al. (2022) model while benefiting from a considerably *simpler* training procedure, with *faster* convergence to *better* solutions. Figure 2 provides an overview of its architecture. In essence, the main difference with TACTiS lies in the choice of the copula-based density estimator (Sec. 3): rather than being *permutation-invariant* (Sec. 3.1), it is of the proposed *two-stage type* (Sec. 3.2). This difference mandates changes to the architecture and the loss of the model, as well as the introduction of a training curriculum, which we outline below. In what follows, we use $\boldsymbol{\theta}$ to denote parameters of neural networks, which are not to be confused with distributional parameters, denoted by $\boldsymbol{\phi}$.

**Dual Encoder:** Whereas TACTiS relies on a *single encoder* to produce all parameters of the output density, TACTiS-2 relies on *two distinct encoders* ($\text{Enc}_{\theta_M}$ and $\text{Enc}_{\theta_C}$) whose representations are used to parametrize the marginal CDFs ($F_{\phi_i}$) and the copula distribution ($c_{\phi_c}$), respectively. As in TACTiS, these are transformer encoders that embed realizations of both observed $\mathbf{X}^{(o)}$ and missing $\mathbf{X}^{(m)}$ values, with the missing values masked as in *masked language models* (Devlin et al., 2018)). Let $x_i$ refer to the realization of a generic random variable (missing or observed) and $m_i$, $\mathbf{c}_i$, and $t_i$ its corresponding mask, covariates, and timestamp. The representations are obtained as:

$$\mathbf{z}_i^M = \text{Enc}_{\theta_M}(x_i \cdot m_i, \mathbf{c}_i, m_i, \mathbf{p}_i), \qquad \mathbf{z}_i^C = \text{Enc}_{\theta_C}(x_i \cdot m_i, \mathbf{c}_i, m_i, \mathbf{p}_i), \tag{9}$$

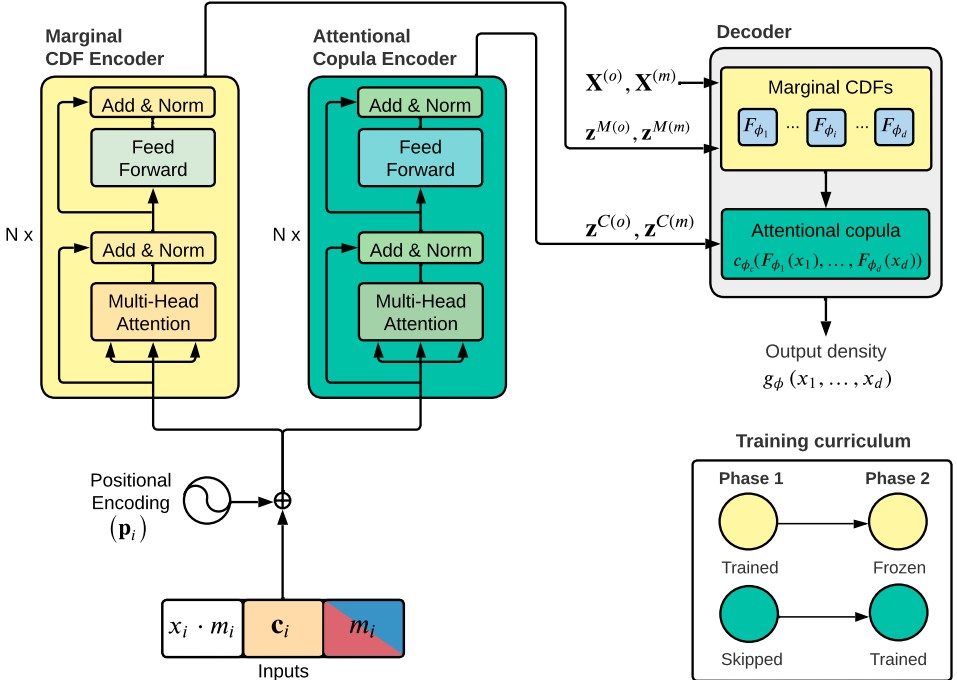

**Figure 2:** *The* TACTiS-2 *architecture with the dual encoder and the decoder. The training curriculum (bottom right) shows the proposed two-stage approach.*

where information about the timestamp $t_i$ is incorporated in the process via an additive positional encoding $\mathbf{p}_i$, which we take to be sinusoidal features as in Vaswani et al. (2017).

**Decoder:** As in TACTiS, the decoder estimates the conditional density of missing values $\mathbf{X}^{(m)}$ (Eq. (1)) with a copula-based estimator structured as in Eq. (3). It consists of two modules, which are tasked with producing the distributional parameters for (i) the marginal CDFs $F_{\phi_i}$, and (ii) the copula $c_{\phi_c}$, respectively. The modules are akin to those used in TACTiS but differ in that they only make use of their respective encodings: $\mathbf{z}^M$ and $\mathbf{z}^C$. For completeness, we outline them below.

**(Marginals)** The first module is a Hyper-Network ($\mathrm{HN}_{\theta_M}$) producing the parameters $\phi_i$ of *Deep Sigmoidal Flows* (DSF) (Huang et al., 2018)[1] that estimate each $F_{\phi_i}$. The value $u_i$, corresponding to the probability integral transform of $x_i$, is then obtained as

$$\phi_i = \mathrm{HN}_{\theta_M}\big(\mathbf{z}_i^M\big), \qquad u_i = \mathrm{DSF}_{\phi_i}(x_i). \tag{10}$$

**(Copula)** The second module parametrizes a construct termed *attentional copula*. This corresponds to a factorization of the copula density, as in Eq. (5), where each conditional is parametrized using a causal attention mechanism (Vaswani et al., 2017). More formally, consider an arbitrary ordering of the missing variables $\mathbf{X}^{(m)}$ and let $c_{\phi_{c,i}}(u_i \mid u_1, \ldots, u_{i-1})$ be the $i$-th term in the factorization. Parameters $\phi_{c,i}$ are produced by attending to the $\mathbf{z}_j^C$ (see Eq. (9)) and $u_j$ (see Eq. (10)) for all observed tokens, denoted by $\mathbf{z}^{C(o)}$ and $\mathbf{u}^{(o)}$, and for all missing variables that precede in the ordering, denoted by $\mathbf{z}_{1:i-1}^{C(m)}$ and $\mathbf{u}_{1:i-1}^{(m)}$:

$$\mathbf{K}_i = \mathrm{Key}_{\theta_C}(\mathbf{z}^{C(o)}, \mathbf{z}_{1:i-1}^{C(m)}, \mathbf{u}^{(o)}, \mathbf{u}_{1:i-1}^{(m)}) \qquad \mathbf{V}_i = \mathrm{Value}_{\theta_C}(\mathbf{z}^{C(o)}, \mathbf{z}_{1:i-1}^{C(m)}, \mathbf{u}^{(o)}, \mathbf{u}_{1:i-1}^{(m)})$$

$$\mathbf{q}_i = \mathrm{Query}_{\theta_C}(\mathbf{z}_i^{C(m)}) \qquad\qquad\qquad \phi_{c,i} = \mathrm{Attn}_{\theta_C}(\mathbf{q}_i, \mathbf{K}_i, \mathbf{V}_i)$$

where $\mathrm{Attn}_{\theta_C}$ is an attention mechanism that attends to the keys $\mathbf{K}_i$ and values $\mathbf{V}_i$ using query $\mathbf{q}_i$, and applies a non-linear transformation to the output. As in Drouin et al. (2022), we take each $c_{\phi_{c,i}}$ to be a histogram distribution with support in $[0, 1]$, but other choices are compatible with this approach.

**Curriculum Learning:** A crucial difference between TACTiS and TACTiS-2 lies in their training procedure. TACTiS optimizes the cumbersome, permutation-based, Problem (6). In contrast, TACTiS-2 is trained by maximum likelihood using a training curriculum, enabled by the use of a *dual encoder*, which we show is equivalent to solving the two-phase Problem (7). See Fig. 2 for an illustration.

---

[1]The original Deep Sigmoidal Flow is modified such that it outputs values in $[0, 1]$.

In the first phase, only the parameters $\theta_M$ for the marginal components are trained, while those for the copula, $\theta_C$, are skipped. This boils down to optimizing Problem (4) using a trivial copula where all variables are independent (which we denote by $c_I$):

$$\arg\min_{\theta_M} - \mathop{\mathbb{E}}_{\mathbf{x} \sim \mathbf{X}} \log \left[ c_I \left( F_{\phi_1}(x_1^{(m)}), \dots, F_{\phi_{n_m}}(x_d^{(m)}) \right) \cdot \prod_{i=1}^{d} f_{\phi_i}(x_i^{(m)}) \right], \qquad (11)$$

where each $f_{\phi_i}$ is obtained by differentiating $F_{\phi_i}$ w.r.t. $x_i$, an operation that is efficient for DSFs. Since by definition $c_I(\dots) \equiv 1$, this problem reduces to Problem (8).

In the second phase, the parameters learned for the marginal components $\theta_M$ are frozen and the parameters for the copula components $\theta_C$ are trained until convergence. The optimization problem is hence given by:

$$\arg\min_{\theta_C} - \mathop{\mathbb{E}}_{\mathbf{x} \sim \mathbf{X}} \log \left[ c_{\phi_c}(F_{\phi_1^\star}(x_1^{(m)}), \dots, F_{\phi_d^\star}(x_d^{(m)})) \cdot \prod_{i=1}^{d} f_{\phi_i^\star}(x_i^{(m)}) \right], \qquad (12)$$

which reduces to Problem (7), since $\{f_{\phi_i^\star}(\dots)\}_{i=1}^{d}$ are constant. Hence, by Prop. 2, we have that, given sufficient capacity, the *attentional copulas* learned by TACTiS-2 will be valid. We stress the importance of the proposed learning curriculum, since by Prop. 1, we know that simply maximizing the likelihood w.r.t. $\theta_M$ and $\theta_C$ is highly unlikely to result in valid copulas.

**Sampling:** Inference proceeds as in TACTiS: (i) sampling according to the copula density and (ii) applying the inverse CDFs to obtain samples from Eq. (1). We defer to Drouin et al. (2022) for details.

## 5 EXPERIMENTS

We start by empirically validating the two-stage approach to learning attentional copulas (Sec. 5.1). Then, we show that TACTiS-2 achieves state-of-the-art performance in a forecasting benchmark and that it can perform highly accurate interpolation (Sec. 5.2). Finally, we show that TACTiS-2 outperforms TACTiS in all aspects, namely accuracy and training dynamics, while preserving its high flexibility.

### 5.1 EMPIRICAL VALIDATION OF TWO-STAGE ATTENTIONAL COPULAS

According to Prop. 2, TACTiS-2 should learn valid copulas. We empirically validate this claim in a setting where the sample size, training time, and capacity are finite. As in Drouin et al. (2022), we rely on an experiment in which data is drawn from a bivariate distribution with a known copula structure. The results in Fig. 3 show that, in this setting, TACTiS-2 recovers a valid copula that closely matches the ground truth. See App. B.3 for experimental details and additional results.

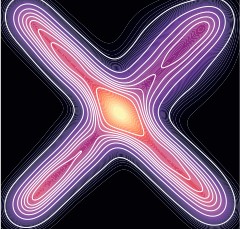

**Figure 3:** *The density of the learned copula (contours) closely matches that of the ground truth (colors).*

### 5.2 EVALUATION OF PREDICTIVE PERFORMANCE

We now evaluate the forecasting and interpolation abilities of TACTiS-2 in a benchmark of five common real-world datasets from the *Monash Time Series Forecasting Repository* (Godahewa et al., 2021): `electricity`, `fred-md`, `kdd-cup`, `solar-10min`, and `traffic`. These were chosen due to their diverse dimensionality ($n \in [107, 826]$), sampling frequencies (monthly, hourly, and 10 min.), and prediction lengths ($\ell_i \in [12, 72]$). All datasets are detailed in App. C.1.

**Evaluation Protocol:** The evaluation follows the protocol of Drouin et al. (2022), which consists in a backtesting procedure that combines rolling-window evaluation with periodic retraining. The estimated distributions are scored using the Continuous Ranked Probability Score (CRPS; Matheson & Winkler, 1976), the CRPS-Sum (Salinas et al., 2019)—a multivariate generalization of the CRPS—and the Energy score (Gneiting & Raftery, 2007), as is standard in the literature. We further compare some models using the negative log-likelihood (NLL), which was found to be more effective in detecting errors in modeling multivariate dependencies (Marcotte et al., 2023). Experimental details are provided in App. C.

**Forecasting Benchmark:** We compare TACTiS-2 with the following state-of-the-art multivariate probabilistic forecasting methods: GPVar (Salinas et al., 2019), an LSTM-based method that parametrizes a Gaussian copula; TempFlow (Rasul et al., 2021b), a method that parametrizes normalizing flows using

**Table 1:** *Mean CRPS-Sum values for the forecasting experiments. Standard errors are calculated using the Newey-West (1987; 1994) estimator. Average ranks report the average ranking of methods across all evaluation windows and datasets. Lower is better and the best results are in bold.*

| Model | electricity | fred-md | kdd-cup | solar-10min | traffic | Avg. Rank |
|---|---|---|---|---|---|---|
| Auto-ARIMA | $0.077 \pm 0.016$ | $0.043 \pm 0.005$ | $0.625 \pm 0.066$ | $0.994 \pm 0.216$ | $0.222 \pm 0.005$ | $6.2 \pm 0.3$ |
| ETS | $0.059 \pm 0.011$ | $0.037 \pm 0.010$ | $0.408 \pm 0.030$ | $0.678 \pm 0.097$ | $0.353 \pm 0.011$ | $6.0 \pm 0.2$ |
| TempFlow | $0.075 \pm 0.024$ | $0.095 \pm 0.004$ | $0.250 \pm 0.010$ | $0.507 \pm 0.034$ | $0.242 \pm 0.020$ | $5.4 \pm 0.2$ |
| SPD | $0.062 \pm 0.016$ | $0.048 \pm 0.011$ | $0.319 \pm 0.013$ | $0.568 \pm 0.061$ | $0.228 \pm 0.013$ | $5.2 \pm 0.3$ |
| TimeGrad | $0.067 \pm 0.028$ | $0.094 \pm 0.030$ | $0.326 \pm 0.024$ | $0.540 \pm 0.044$ | $0.126 \pm 0.019$ | $5.0 \pm 0.2$ |
| GPVar | $0.035 \pm 0.011$ | $0.067 \pm 0.008$ | $0.290 \pm 0.005$ | $0.254 \pm 0.028$ | $0.145 \pm 0.010$ | $3.8 \pm 0.2$ |
| TACTiS | $0.021 \pm 0.005$ | $0.042 \pm 0.009$ | $0.237 \pm 0.013$ | $0.311 \pm 0.061$ | $\mathbf{0.071 \pm 0.008}$ | $2.4 \pm 0.2$ |
| TACTiS-2 | $\mathbf{0.020 \pm 0.005}$ | $\mathbf{0.035 \pm 0.005}$ | $\mathbf{0.234 \pm 0.011}$ | $\mathbf{0.240 \pm 0.027}$ | $0.078 \pm 0.008$ | $\mathbf{1.9 \pm 0.2}$ |

**Table 2:** *Mean NLL values for forecasting experiments and training FLOP counts. Standard errors are calculated using the Newey-West (1987; 1994) estimator. Lower is better. Best results in bold. .*

| Model | | electricity | fred-md | kdd-cup | solar-10min | traffic |
|---|---|---|---|---|---|---|
| TACTiS | NLL | $11.028 \pm 3.616$ | $1.364 \pm 0.253$ | $2.281 \pm 0.770$ | $-2.572 \pm 0.093$ | $1.249 \pm 0.080$ |
| | FLOPs ($\times 10^{16}$) | $1.931 \pm 0.182$ | $1.956 \pm 0.192$ | $1.952 \pm 0.208$ | $0.174 \pm 0.018$ | $1.207 \pm 0.517$ |
| TACTiS-2 | NLL | $\mathbf{10.674 \pm 2.867}$ | $\mathbf{0.378 \pm 0.076}$ | $\mathbf{1.055 \pm 0.713}$ | $\mathbf{-4.333 \pm 0.181}$ | $\mathbf{-0.358 \pm 0.077}$ |
| | FLOPs ($\times 10^{16}$) | $\mathbf{0.623 \pm 0.018}$ | $\mathbf{0.738 \pm 0.022}$ | $\mathbf{0.324 \pm 0.014}$ | $\mathbf{0.078 \pm 0.005}$ | $\mathbf{0.289 \pm 0.061}$ |

transformers; TimeGrad (Rasul et al., 2021a), an autoregressive model based on denoising diffusion; and Stochastic Process Diffusion (SPD) (Biloš et al., 2023), the only other general-purpose approach which models time series as continuous functions using stochastic processes as noise sources for diffusion. Moreover, we include the following classical forecasting methods, which tend to be strong baselines (Makridakis et al., 2018a;b; 2022): ARIMA (Box et al., 2015) and ETS exponential smoothing (Hyndman et al., 2008).

The CRPS-Sum results are reported in Tab. 1. Clearly, TACTiS-2 shows state-of-the-art performance, achieving the lowest values on 4 out of 5 datasets, while being slightly outperformed by TACTiS on traffic. Looking at the average rankings, which are a good indicator of performance as a general-purpose forecasting tool, we see that TACTiS-2 outperforms all baselines, with TACTiS being its closest competitor. To further contrast these two, we perform a comparison of NLL and report the results in Tab. 2 and Fig. 1. We observe that TACTiS-2 outperforms TACTiS, additionally with no overlap in the confidence intervals for 3 out of 5 datasets, including traffic. This is strong evidence that TACTiS-2 better captures multivariate dependencies (Marcotte et al., 2023), which is plausible given the proposed improvements to the attentional copula. The results for all other metrics are in line with the above findings and are reported in App. A.2.

**Training Dynamics:**    As discussed in Sec. 3, the optimization problem solved by TACTiS-2 is considerably simpler than the one solved by TACTiS. We quantify this by measuring the number of floating-point operations (FLOPs) required to train until convergence in the forecasting benchmark (see Tab. 2). From these results, it is clear that TACTiS-2 achieves greater accuracies while using much less compute than TACTiS. As additional evidence of improved training dynamics, we report training curves that compare the NLL, on a validation set, with respect to training FLOPs for TACTiS-2, TACTiS, and an ablation of TACTiS-2 that does not use the two-stage curriculum. Results for the kdd-cup dataset are reported in Fig. 4 and those for other datasets are available in App. A.3. From these, we reach two conclusions: (i) TACTiS-2 converges much faster to better solutions, (ii) the two-stage curriculum is crucial TACTiS-2's success.

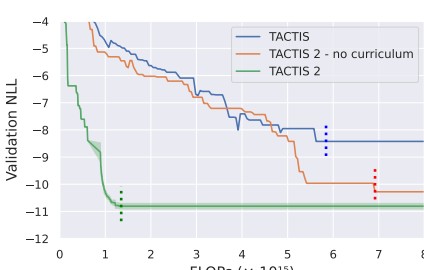

**Figure 4:** *TACTiS-2 converges to better NLLs using fewer FLOPs than TACTiS, as well as an ablation that trains all parameters jointly without the two-stage curriculum. Vertical bars indicate the latest convergence point over 5 runs with a maximum duration of three days.*

**Interpolation Performance:**    While both TACTiS and TACTiS-2 are capable of interpolation, the results reported in Tab. 3 indicate that TACTiS-2 is much better at this task. This is illustrated in Fig. 5a, which

**Table 3:** *Mean NLL values for the interpolation experiments. Standard errors are calculated using the Newey-West (1987; 1994) estimator. Lower is better and the best results are in bold.*

| Model | electricity | fred-md | kdd-cup | solar-10min | traffic |
|---|---|---|---|---|---|
| TACTiS | $-0.159 \pm 0.007$ | $-0.119 \pm 0.021$ | $1.050 \pm 0.045$ | $-0.948 \pm 0.089$ | $0.825 \pm 0.149$ |
| TACTiS-2 | $\mathbf{-0.189 \pm 0.034}$ | $\mathbf{-0.250 \pm 0.017}$ | $\mathbf{0.138 \pm 0.043}$ | $\mathbf{-3.262 \pm 0.186}$ | $\mathbf{-0.431 \pm 0.063}$ |

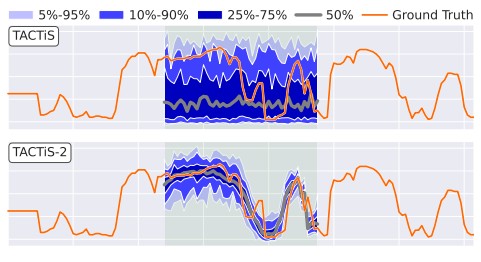

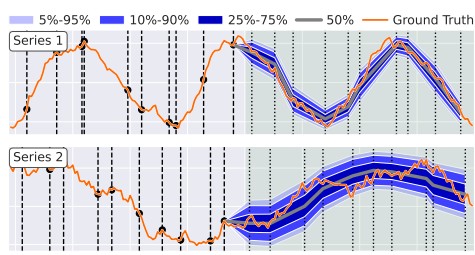

**(a)** *Interpolation:* TACTiS-2 *(bottom) produces more accurate interpolation distributions than* TACTiS *(top) – example from* kdd-cup.

**(b)** TACTiS-2 *correctly forecasts two unaligned/unevenly-sampled series on the noisy sines dataset. Dashed bars indicate measurements.*

**Figure 5:** *An illustration of the flexibility of* TACTiS-2.

shows an example in which TACTiS-2 produces a much more plausible interpolation distribution than TACTiS. Additional examples are available in App. A.5.

**Model Flexibility:** Like TACTiS, the TACTiS-2 architecture supports heterogenous datasets composed of unaligned series with uneven sampling frequencies. We illustrate this by replicating an experiment of Drouin et al. (2022) which consists in forecasting a bivariate noisy sine process with irregularly-spaced observations. The results in Fig. 5b show that TACTiS-2 faithfully performs forecasting in this setting, preserving the flexiblity of TACTiS. Additional results on real-world datasets are available in App. D.1.

## 6 RELATED WORK

**Deep Learning for Probabilistic Time Series Prediction:** The majority of prior work in this field has been geared toward forecasting tasks. Early work explored the application of recurrent and convolutional neural networks to estimating univariate forecast distributions (Rangapuram et al., 2018; Shih et al., 2019; Chen et al., 2020; de Bézenac et al., 2020; Yanchenko & Mukherjee, 2020). More relevant to the present work are methods for *multivariate* forecasting, which we review in light of the desiderata outlined in Sec. 1. Many such works lack the flexibility to model arbitrary joint distributions. For instance, DeepAR (Salinas et al., 2020) uses an RNN to parametrize an autoregressive factorization of the joint density using a chosen parametric form (e.g., Gaussian). Similarly, GPVar (Salinas et al., 2019) uses an RNN to parametrize a low-rank Gaussian copula approximation of the joint distribution. Wu et al. (2020) rely on an adversarial sparse transformer to estimate conditional quantiles of the predictive distribution but do not model the full joint density. Other approaches relax such limitations, by enabling the estimation of arbitrary distributions. For instance, TempFlow (Rasul et al., 2021b) uses RNNs and transformers (Vaswani et al., 2017) to parametrize a multivariate normalizing flow (Papamakarios et al., 2021). TimeGrad (Rasul et al., 2021a) models the one-step-ahead predictive joint distribution with a denoising diffusion model (Ho et al., 2020; Sohl-Dickstein et al., 2015) conditioned on an RNN. However, all of the above lack the flexibility to handle unaligned/unevenly-sampled series, missing data, and are limited to forecasting.

Recently, multiple works have made progress toward our desiderata. Among those, TACTiS (Drouin et al., 2022), which we have extensively described, and SPD (Biloš et al., 2023), which uses stochastic processes as noise sources during diffusion, are the only ones that satisfy all of them. An alternative approach, CSDI (Tashiro et al., 2021), which relies on a conditional score-based diffusion model trained using a self-supervised interpolation objective, supports both forecasting and interpolation. However, it cannot learn from unaligned time series. Similarly, SSSD (Alcaraz & Strodthoff, 2023), a conditional diffusion model that relies on structured state-space models (Gu et al., 2022) as internal layers, can perform both forecasting and interpolation but does not support unaligned and unevenly-sampled time series.

**Copulas for Probabilistic Forecasting:** Copula-based models for multivariate forecasting have been extensively studied in economics and finance (Patton, 2012; Größer & Okhrin, 2021), with applications such as modeling financial returns and volatility (Bouyé & Salmon, 2009; Bouyé et al., 2008; Wang & Tao,

2020). A common semiparametric approach consists of combining a parametric copula, e.g., Archimedean, with nonparametric empirical estimates of the marginal CDFs (ECDFs). One notable example is the work of Salinas et al. (2019), which combines ECDFs with an RNN that dynamically parametrizes a Gaussian copula. A key limitation of such approaches is that ECDF estimates are only valid for stationary processes. Moving away from this assumption, Wen & Torkkola (2019) propose to use neural networks to model the CDFs in addition to a Gaussian copula. However, the choice of a Gaussian copula is still a strong parametric assumption which we seek to avoid. Closer to our work are methods that relax such parametric assumptions, such as Krupskii & Joe (2020); Mayer & Wied (2021); Toubeau et al. (2019); Drouin et al. (2022). Toubeau et al. (2019) use the historical data to estimate a histogram-based nonparametric copula. A key caveat to this approach is that it assumes that the historical data is independent and identically distributed. Finally, the work most similar to ours is the nonparametric approach of Drouin et al. (2022), which, as extensively described, we significantly simplify and improve upon.

**Copulas in Machine Learning:** Beyond time series, copulas have been applied to various machine learning problems, such as domain adaptation (Lopez-Paz et al., 2012), variational inference (Tran et al., 2015; Hirt et al., 2019), learning disentangled representations (Wieser et al., 2018), dependency-seeking clustering (Rey & Roth, 2012), and generative modeling (Sexton et al., 2022; Tagasovska et al., 2019; Wang & Wang, 2019). Hence, we emphasize that our proposed transformer-based nonparametric copulas are applicable beyond time series and review two closely related works. First, Janke et al. (2021) propose an approach to learning nonparametric copula that uses a generative adversarial network (Goodfellow et al., 2014) to learn a latent distribution on the unit cube. This distribution is then transformed into a valid copula distribution using its ECDFs. As opposed to our approach, which is fully differentiable, their reliance on ECDFs leads to a non-differentiable objective that must be approximated during training. Second, Wiese et al. (2019) use normalizing flows to parametrize both the marginals and the copulas, resulting in an approach that is fully differentiable. However, they focus on the bivariate case and rely on vine copulas to extend their approach to multivariate copulas, something that our approach does not require.

## 7 DISCUSSION

This work introduces TACTiS-2, a general-purpose model for multivariate probabilistic time series prediction, combining the flexibility of transformers with a new approach to learning attention-based nonparametric copulas. TACTiS-2 establishes itself as the new state-of-the-art model for forecasting on several real-world datasets, while showing better training dynamics than its predecessor, TACTiS. This superior performance is mainly due to its simplified optimization procedure, which ultimately allows it to reach better solutions, in particular better copulas. TACTiS-2's performance is further enhanced by the use of a dual-encoder that learns representations specialized for each distributional component.

There are multiple ways in which TACTiS-2 could be improved. For example, it would be interesting to incorporate inductive biases specific to time series data into the architecture, e.g., using Fourier features to learn high-frequency patterns (Woo et al., 2022), and auto-correlation mechanisms in the attention layers (Wu et al., 2021). Next, TACTiS-2 is limited to working with continuous data due to its usage of normalizing flows for the marginal distributions. This limitation could be addressed by building on prior work (Tran et al., 2019; Ziegler & Rush, 2019) to adapt the normalizing flows to work with discrete distributions. Finally, it is important to note that, as for all related work (Wiese et al., 2019; Drouin et al., 2022; Janke et al., 2021), the nonparametric copulas learned by TACTiS-2 are valid in the limit of infinite data and capacity. As future work, it would be interesting to study the convergence properties of these approaches to valid copulas in settings with finite samples and non-convex optimization landscapes.

Beyond the scope of this work, there are interesting settings in which the capabilities of TACTiS-2 could be studied further. The proposed form of fully decoupling the marginal distributions and the dependency structure could be especially useful in handling distribution shifts, which are common in real-world time series (Yao et al., 2022; Gagnon-Audet et al., 2023). Such a decoupling allows changes in either the marginal distributions or the dependency structure or both these factors to be understood separately in scenarios of distribution shift, allowing to adapt the model appropriately. Next, it would be interesting to study large-scale training of TACTiS-2 on several related time series from a specific domain, where marginal distributions can be trained for each series, while the attentional copula component can be shared. Finally, it would be interesting to exploit the flexibility of TACTiS-2 for multitask pretraining, where the model is jointly trained on multiple probabilistic prediction tasks such as forecasting, interpolation and imputation, and can be used as a general-purpose model downstream. Such extensions to TACTiS-2 constitute exciting directions towards foundation models (Bommasani et al., 2021) for time series.

## ACKNOWLEDGEMENTS

The authors are grateful to Putra Manggala, Sébastien Paquet, and Sokhna Diarra Mbacke for their valuable feedback and suggestions. The authors are grateful to Juan Rodriguez and Daniel Tremblay for their help with the experiments. This research was supported by a Mitacs Accelerate Grant.

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

# Appendix

## Table of Contents

## A  METRICS AND ADDITIONAL RESULTS

### A.1  METRIC DEFINITIONS

**CRPS:** The Continuous Ranked Probability Score (Matheson & Winkler, 1976; Gneiting & Raftery, 2007) is a univariate metric used to assess the accuracy of a forecast distribution $X$, given a ground-truth realization $x^*$:

$$\text{CRPS}(x^*, X) \stackrel{\text{def}}{=} 2 \int_0^1 \left( \mathbf{1}\big[ F^{-1}(q) - x^* \big] - q \right) \left( F^{-1}(q) - x^* \right) dq, \tag{13}$$

where $F(x)$ is the cumulative distribution function (CDF) of the forecast distribution, and $\mathbf{1}[x]$ is the Heaviside function. We follow previous authors in normalizing the CRPS results using the mean absolute value of the ground-truth realizations of each series, before taking the average over all series and time steps. Since the CRPS is a univariate metric, it fully ignores the quality of the multivariate dependencies in the forecasts being assessed.

**CRPS-Sum:** The CRPS-Sum (Salinas et al., 2019) is an adaptation of the CRPS to take into account some of the multivariate dependencies in the forecasts. The CRPS-Sum is the result of summing both the forecast and the ground-truth values over all series, before computing the CRPS over the resulting sums.

**Energy Score:** The Energy Score (ES) (Gneiting & Raftery, 2007) is a multivariate metric. For a multivariate forecast distribution $\mathbf{X}$ and ground-truth realisation $\mathbf{x}^*$, it is computed as:

$$\text{ES}(\mathbf{x}^*, \mathbf{X}) \stackrel{\text{def}}{=} \mathop{\mathbb{E}}_{\mathbf{x} \sim \mathbf{X}} |\mathbf{x} - \mathbf{x}^*|_2^{\beta} - \frac{1}{2} \mathop{\mathbb{E}}_{\substack{\mathbf{x} \sim \mathbf{X} \\ \mathbf{x}' \sim \mathbf{X}}} |\mathbf{x} - \mathbf{x}'|_2^{\beta}, \tag{14}$$

where $|\mathbf{x}|_2$ is the Euclidean norm and $0 < \beta \le 2$ is a parameter we set to 1.

**NLL:** The Negative Log-Likelihood (NLL) is a metric that directly uses the forecast probability distribution function (PDF) instead of sampling from it. In the case of a continuous distribution with PDF $p(\mathbf{x})$ and a ground-truth realisation $\mathbf{x}^*$, the NLL is as follow:

$$\text{NLL}(\mathbf{x}^*, p(\mathbf{x})) \stackrel{\text{def}}{=} -\log p(\mathbf{x}^*). \tag{15}$$

We normalize the NLL results by dividing it by the number of dimensions of the forecast.

**Standard Error Computation:** In all of our metric results, the standard errors (indicated by $\pm$) are computed using the Newey-West (1987; 1994) estimator. This estimator takes into account the autocorrelation and heteroscedasticity that are inherent to the sequential nature of our backtesting procedure, leading to wider standard errors than those computed using the assumption that the metric values for each backtesting period are independent. In particular, we use the implementation from the R `sandwich` package (Zeileis et al., 2020), with the Bartlett kernel weights, 3 lags, and automatic bandwidth selection.

## A.2 RESULTS ON ALTERNATIVE FORECASTING METRICS

**Table 4:** *CRPS means: Averaged across all backtest sets. Standard errors are calculated using the Newey-West (1987; 1994) estimator. Average ranks report the average ranking of methods across all evaluation windows and datasets. Lower is better and the best results are in bold.*

| Model | electricity | fred-md | kdd-cup | solar-10min | traffic | Avg. Rank |
|---|---|---|---|---|---|---|
| ETS | $0.094 \pm 0.014$ | $0.050 \pm 0.011$ | $0.560 \pm 0.028$ | $0.844 \pm 0.119$ | $0.437 \pm 0.012$ | $6.5 \pm 0.2$ |
| Auto-ARIMA | $0.129 \pm 0.015$ | $0.052 \pm 0.005$ | $0.477 \pm 0.015$ | $0.636 \pm 0.060$ | $0.310 \pm 0.004$ | $5.7 \pm 0.2$ |
| TempFlow | $0.109 \pm 0.024$ | $0.110 \pm 0.003$ | $0.451 \pm 0.005$ | $0.547 \pm 0.036$ | $0.320 \pm 0.015$ | $5.6 \pm 0.2$ |
| TimeGrad | $0.101 \pm 0.027$ | $0.142 \pm 0.058$ | $0.495 \pm 0.023$ | $0.560 \pm 0.047$ | $0.217 \pm 0.015$ | $5.2 \pm 0.3$ |
| SPD | $0.099 \pm 0.016$ | $0.058 \pm 0.011$ | $0.465 \pm 0.005$ | $0.585 \pm 0.050$ | $0.283 \pm 0.007$ | $5.1 \pm 0.3$ |
| GPVar | $0.067 \pm 0.010$ | $0.086 \pm 0.009$ | $0.459 \pm 0.009$ | $0.298 \pm 0.034$ | $0.213 \pm 0.009$ | $4.1 \pm 0.1$ |
| TACTiS | $0.052 \pm 0.006$ | $0.048 \pm 0.010$ | $0.420 \pm 0.007$ | $0.326 \pm 0.049$ | $\mathbf{0.161 \pm 0.009}$ | $2.2 \pm 0.1$ |
| TACTiS-2 | $\mathbf{0.049 \pm 0.006}$ | $\mathbf{0.043 \pm 0.006}$ | $\mathbf{0.413 \pm 0.007}$ | $\mathbf{0.256 \pm 0.029}$ | $0.162 \pm 0.010$ | $\mathbf{1.6 \pm 0.2}$ |

**Table 5:** *Energy score means: Averaged across all backtest sets. Standard errors are calculated using the Newey-West (1987; 1994) estimator. Average ranks report the average ranking of methods across all evaluation windows and datasets. Lower is better and the best results are in bold.*

| Model | electricity $\times 10^4$ | fred-md $\times 10^5$ | kdd-cup $\times 10^3$ | solar-10min $\times 10^2$ | traffic $\times 10^0$ | Avg. Rank |
|---|---|---|---|---|---|---|
| Auto-ARIMA | $44.59 \pm 8.56$ | $8.72 \pm 0.81$ | $18.76 \pm 3.31$ | $19.42 \pm 3.37$ | $4.10 \pm 0.05$ | $6.7 \pm 0.2$ |
| TempFlow | $10.25 \pm 2.03$ | $20.16 \pm 0.74$ | $3.30 \pm 0.28$ | $4.25 \pm 0.16$ | $4.59 \pm 0.25$ | $6.1 \pm 0.3$ |
| ETS | $7.94 \pm 0.93$ | $7.90 \pm 1.88$ | $3.60 \pm 0.24$ | $4.74 \pm 0.17$ | $4.98 \pm 0.07$ | $5.7 \pm 0.2$ |
| TimeGrad | $9.69 \pm 2.62$ | $19.87 \pm 7.23$ | $3.30 \pm 0.19$ | $4.31 \pm 0.23$ | $3.38 \pm 0.11$ | $4.7 \pm 0.3$ |
| SPD | $8.90 \pm 1.18$ | $8.94 \pm 1.82$ | $3.13 \pm 0.24$ | $3.68 \pm 0.31$ | $3.94 \pm 0.10$ | $4.4 \pm 0.3$ |
| GPVar | $6.80 \pm 0.62$ | $11.43 \pm 1.60$ | $3.18 \pm 0.20$ | $2.60 \pm 0.10$ | $3.57 \pm 0.10$ | $3.9 \pm 0.2$ |
| TACTiS | $5.42 \pm 0.57$ | $8.18 \pm 1.83$ | $2.93 \pm 0.22$ | $2.88 \pm 0.23$ | $\mathbf{3.10 \pm 0.13}$ | $2.7 \pm 0.3$ |
| TACTiS-2 | $\mathbf{4.91 \pm 0.52}$ | $\mathbf{6.72 \pm 0.10}$ | $\mathbf{2.81 \pm 0.19}$ | $\mathbf{2.37 \pm 0.14}$ | $3.36 \pm 0.27$ | $\mathbf{1.9 \pm 0.2}$ |

## A.3 TRAINING DYNAMICS

We now look deeper into the training dynamics of TACTiS-2. Figs. 6 and 7 compare the validation NLL with respect to the number of FLOPs performed during training on the `kdd-cup` and `solar-10min` datasets respectively. We present such an analysis for TACTiS, TACTiS-2, and we further consider an ablation of TACTiS-2 where the proposed two-stage curriculum (see Sec. 4) is not used. Instead, all the model parameters are trained jointly to minimize the NLL, in a single stage. In doing so, the resulting attentional copulas are absolutely not guaranteed to be valid (Prop. 1). Such an ablation study allows us to understand if the increased performance and efficiency come from the architecture of TACTiS-2 or the two-stage curriculum itself.

From the results, it is clear that TACTiS-2 converges in much fewer FLOPs and to better likelihoods, compared to TACTiS. Notably, TACTiS-2 achieves better performance at any given FLOP budget. TACTiS-2 without the curriculum goes to slightly better negative-log-likelihoods than TACTiS, but requires more FLOPs to reach convergence. This suggests that, in addition to producing valid copulas, the modular architecture of TACTiS-2 and its two-stage curriculum, help it achieve better performance and efficiency.

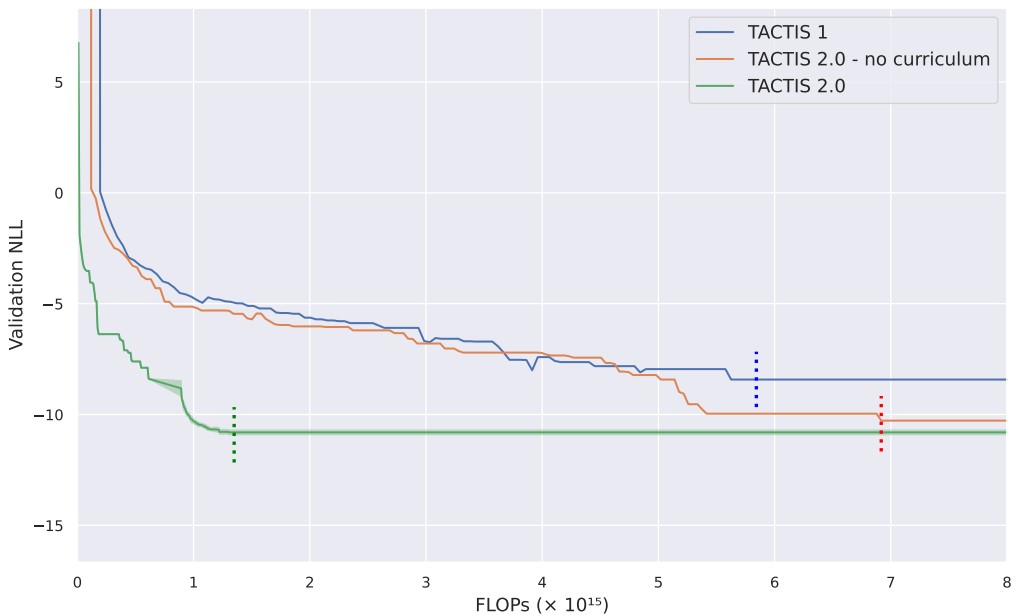

**Figure 6:** *FLOPs consumed vs validation negative log-likelihood on the* kdd-cup *dataset. The results are means of 5 seeds; the shaded region shows the standard error between the seeds. The dashed vertical lines on each curve indicate the latest point of convergence over the 5 seeds with a maximum training duration of three days.*

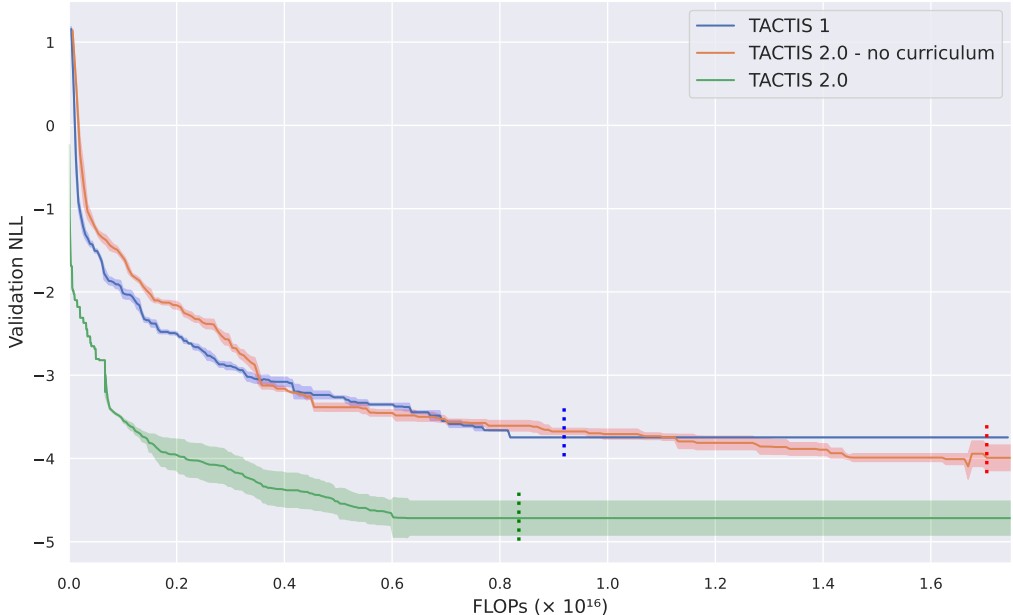

**Figure 7:** *FLOPs consumed vs validation negative log-likelihood on the* solar-10min *dataset. The results are means of 5 seeds; the shaded region shows the standard error between the 5 seeds. The dashed vertical lines on each curve indicate the latest point of convergence over the 5 seeds with a maximum training duration of three days.*

## A.4 Qualitative Results on Forecasting

Figs. 8 and 9 show a few examples forecasts by TACTiS-2 on the `solar-10min` and `electricity` datasets, respectively. These were picked as examples of particularly good forecasts.

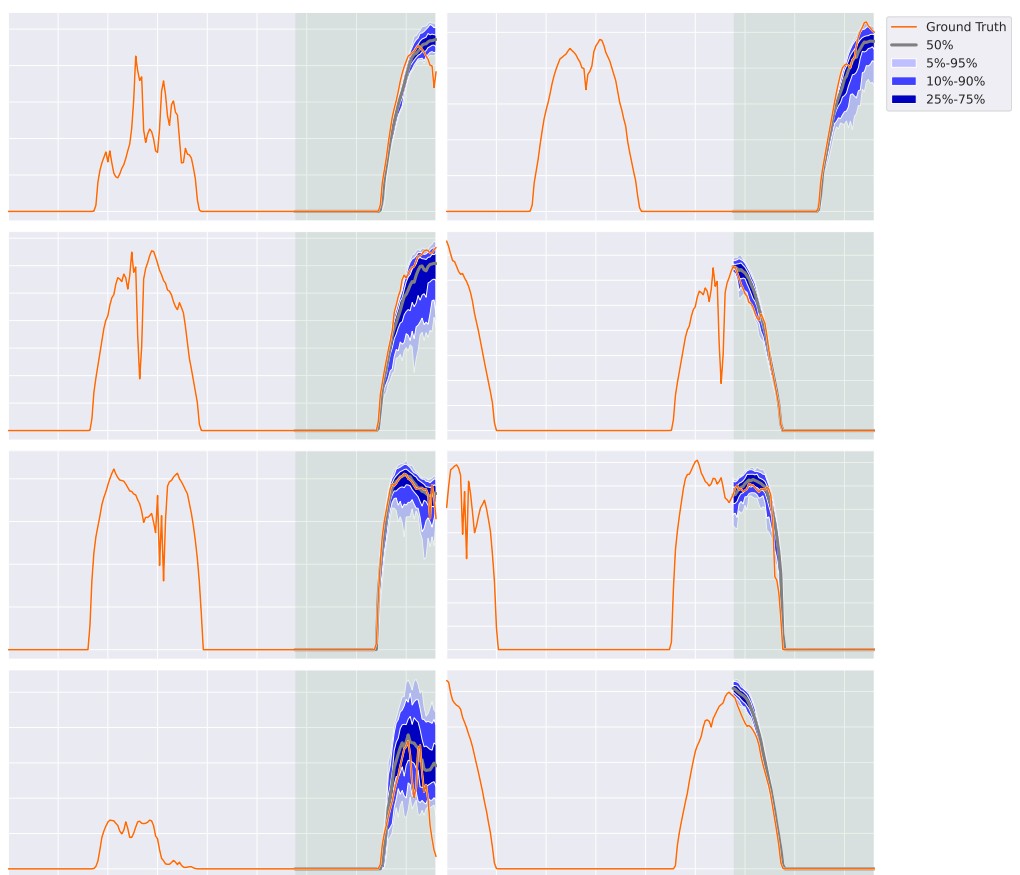

**Figure 8:** *Example forecasts by* TACTiS-2 *on the* `solar-10min` *dataset, along with the historical ground truth.*

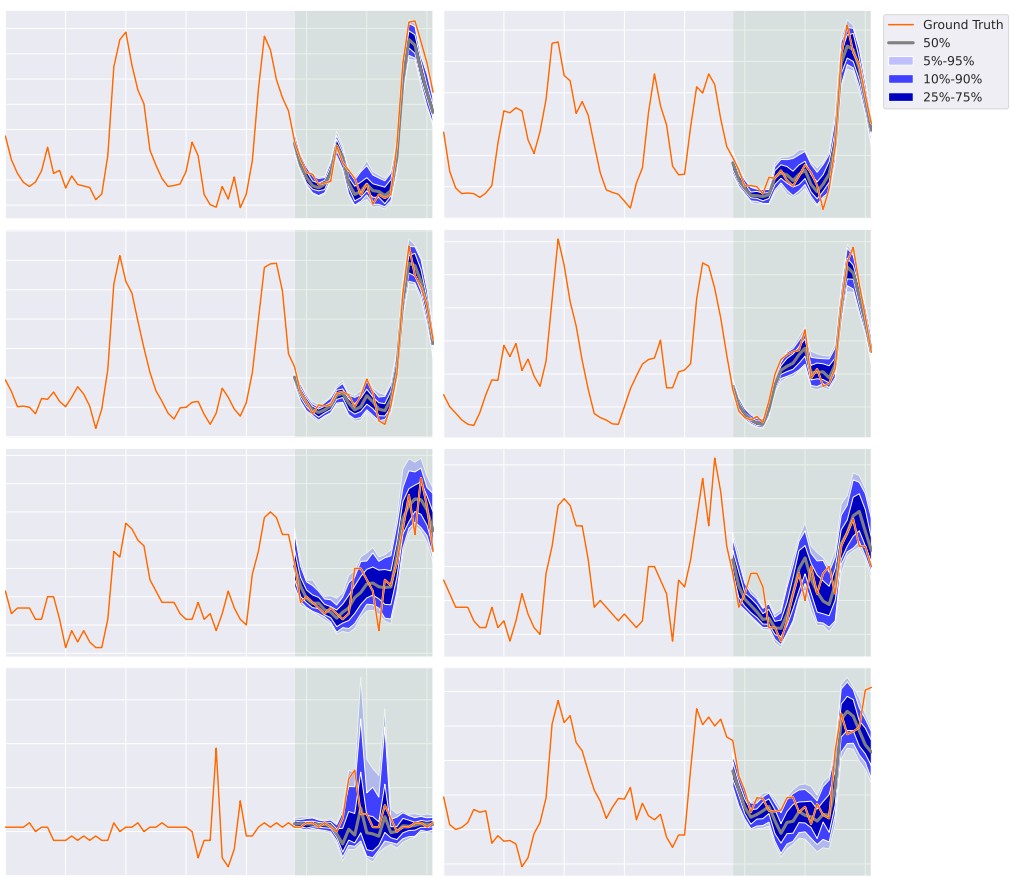

**Figure 9:** *Example forecasts by* TACT*i*S-2 *on the* `electricity` *dataset, along with the historical ground truth.*

## A.5  QUALITATIVE RESULTS ON INTERPOLATION

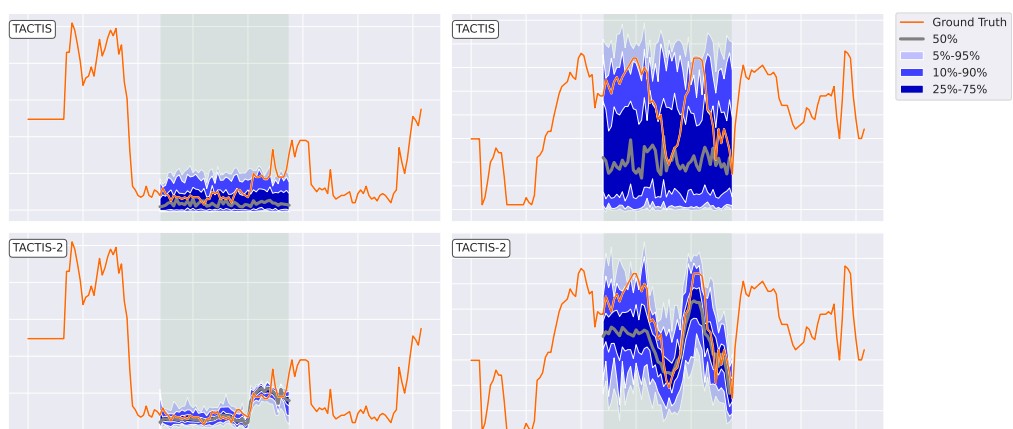

**Figure 10:** *Example interpolations by* TACT*i*S *(top) and* TACT*i*S-2 *(bottom) on the* `kdd-cup` *dataset, along with the context provided to the model.*

Fig. 10 and Fig. 11 show a few examples of interpolations for the `kdd-cup` dataset, comparing TACT*i*S and TACT*i*S-2. Fig. 12 and Fig. 13 show a few examples of interpolations for the `solar-10min` dataset, comparing TACT*i*S and TACT*i*S-2. All these examples were picked randomly.

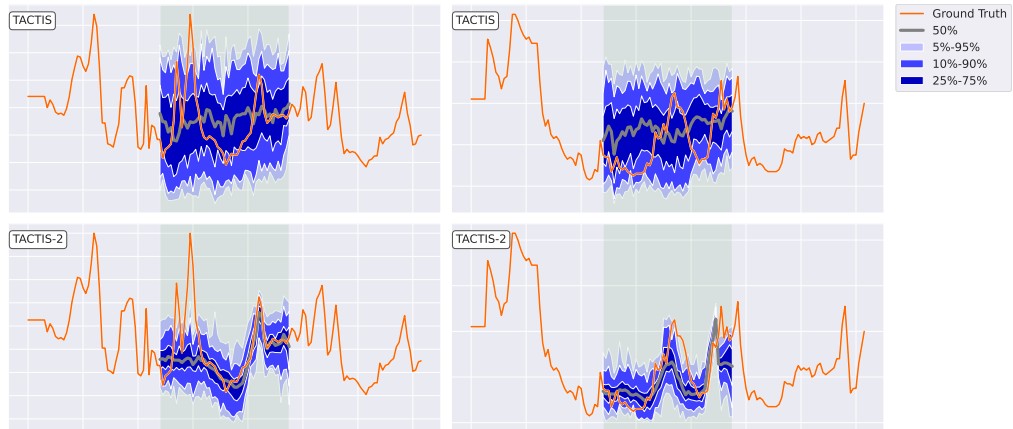

**Figure 11:** *More example interpolations by* TACT*i*S *(top) and* TACT*i*S*-2 (bottom) on the* `kdd-cup` *dataset, along with the context provided to the model.*

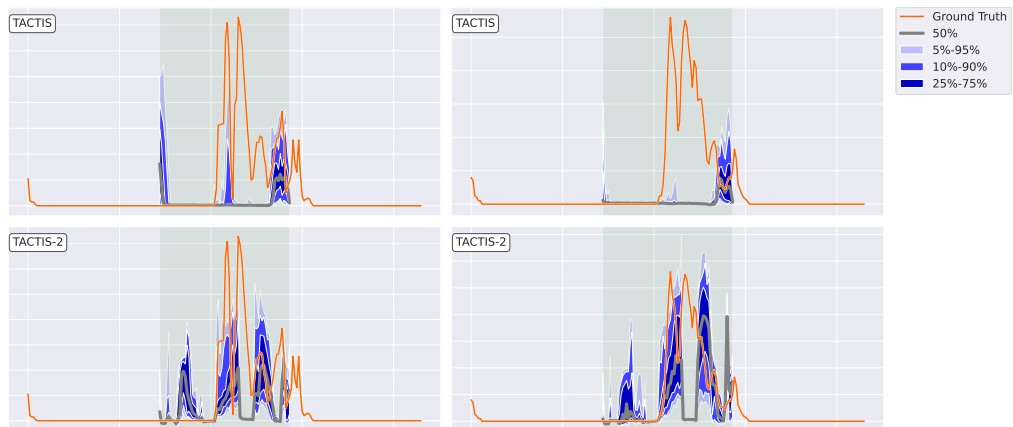

**Figure 12:** *Example interpolations by* TACT*i*S *(top) and* TACT*i*S*-2 (bottom) on the* `solar-10min` *dataset, along with the context provided to the model.*

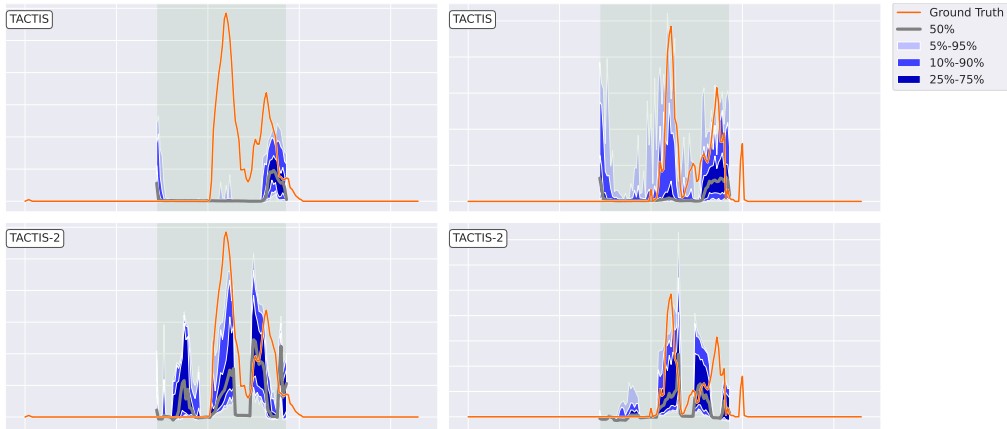

**Figure 13:** *More example interpolations by* TACT*i*S *(top) and* TACT*i*S*-2 (bottom) on the* `solar-10min` *dataset, along with the context provided to the model.*

# B  THEORY AND PROOFS

## B.1  PROOF OF PROPOSITION 1

**Proposition 1**. (Invalid Solutions) Assuming that all random variables $X_1, \ldots, X_d$ have continuous marginal distributions and assuming infinite expressivity for $\{F_{\phi_i}\}_{i=1}^d$ and $c_{\phi_c}$, Problem (4) has infinitely many invalid solutions wherein $c_{\phi_c}$ is not the density function of a valid copula.

*Proof.* Consider a joint distribution over $n$ continuous random variables $X_1, \ldots, X_n$ with CDF $P(x_1, \ldots, x_n)$. According to Sklar's theorem (Sklar, 1959), we know that the joint CDF factorizes as:

$$P(x_1, \ldots, x_n) = C^\star \left( F_1^\star(x_1), \ldots, F_n^\star(x_n) \right),$$

where $C^\star$ and $F_i^\star$ are the true underlying copula and marginal distributions of $P$, respectively. Further, since all $X_i$ are continuous, this factorization is unique.

Now, consider Problem (4). Without loss of generality, the proof considers the joint CDF instead of the PDF that appears in the main text. This problem can be solved by finding $\phi_1, \ldots, \phi_d, \phi_c$, such that $P(X_1, \ldots, X_d) = C_{\phi_c} \left( F_{\phi_1}(x_1) \ldots, F_{\phi_n}(x_n) \right)$. In what follows, we show that, assuming infinite expressivity for $C_{\phi_c}$ and the $F_{\phi_i}$, there are infinitely such solutions where $C_{\phi_c}$ is not a valid copula.

Let $F_{\phi_i} : \mathbb{R} \to [0, 1]$ be an arbitrary strictly monotonic increasing function from the real line to the unit interval. The ground truth CDFs $F_i^\star$ are an example of such functions, but infinitely many alternatives exist, e.g., $F_{\phi_i}(x_i) = \text{sigmoid}(\alpha \cdot x_i)$, where $\alpha \in \mathbb{R}^{>0}$. Then, given any such parametrization of $F_{\phi_i}$ and assuming infinite expressivity for $C_{\phi_c}$, take

$$C_{\phi_c}(u_1, \ldots, u_n) = C^\star \left( F_1^\star(F_{\phi_1}^{-1}(u_1)), \ldots, F_n^\star(F_{\phi_n}^{-1}(u_n)) \right),$$

Due to the invertibility of strictly monotonic functions, this yields a valid solution for Problem (4):

$$C_{\phi_c} \left( F_{\phi_1}(x_1) \ldots, F_{\phi_n}(x_n) \right) = C^\star \left( F_1^\star(x_1), \ldots, F_n^\star(x_n) \right) = P(x_1, \ldots, x_n).$$

Let us now proceed by contradiction to show that, in general, such a $C_{\phi_c}$ is not a valid copula. Assume that $C_{\phi_c}$ is a valid copula and, without loss of generality, that $F_{\phi_1} \neq F_1^\star$. Then, by definition, all univariate marginals of $C_{\phi_c}$ should be standard uniform distributions and thus,

$$C_{\phi_c} \left( 1, \ldots, u_i, \ldots, 1 \right) = u_i.$$

Therefore, we have that:

$$u_1 = C_{\phi_c} \left( u_1, 1, \ldots, 1 \right) = C^\star \left( F_1^\star(F_{\phi_1}^{-1}(u_1)), 1, \ldots, 1 \right) = F_1^\star(F_{\phi_1}^{-1}(u_1)),$$

since $C^\star$ is also a valid copula. However, we know that $F_{\phi_1} \neq F_1^\star$ and thus we arrive at a contradiction.

Hence, without further constraints on $F_{\phi_i}$ and $C_{\phi_c}$, such as the approaches presented in Secs. 3.1 and 3.2, one can obtain arbitrarily many solutions to Problem (4), where $C_{\phi_c}$ is not a valid copula.

$\square$

## B.2  PROOF OF PROPOSITION 2

**Proposition 2**. (Validity) Assuming that all random variables $X_1, \ldots, X_d$ have continuous marginal distributions and assuming infinite expressivity for $\{F_{\phi_i}\}_{i=1}^d$ and $c_{\phi_c}$, solving Problem (7) yields a solution to Problem (4) where $c_{\phi_c}$ is a valid copula.

*Proof.* Let $p(x_1, \ldots, x_d)$ be the joint density of $\mathbf{X} = [X_1, \ldots, X_d]$. According to Sklar (1959), we know that the density can be written as:

$$p(x_1, \ldots, x_d) = c_{\phi_c^\star} \left( F_{\phi_1^\star}(x_1), \ldots, F_{\phi_d^\star}(x_d) \right) \times f_{\phi_1^\star}(x_1) \times \cdots \times f_{\phi_d^\star}(x_d), \tag{16}$$

where $F_{\phi_i^\star}$ and $f_{\phi_i^\star}$ are the ground truth marginal CDF and PDF of $X_i$, respectively, and $c_{\phi_c^\star}$ is the PDF of the ground truth copula. Further, since all $F_{\phi_i^\star}$ are continuous, we know that $c_{\phi_c^\star}$ is unique.

The proof proceeds in two parts: (i) showing that the marginals learned by solving Problem (8) will match the ground truth $(F_{\phi_i^\star}, f_{\phi_i^\star})$, and (ii) showing that, given those true marginals, solving Problem (7) will lead to a copula that matches $c_{\phi_c^\star}$, which is valid by definition.

**Part 1 (Marginals):** Let us recall the nature of Problem (8):

$$\underset{\phi_1,\ldots,\phi_d}{\arg\min} \; - \underset{\mathbf{x}\sim\mathbf{X}}{\mathbb{E}} \log \prod_{i=1}^d f_{\phi_i}(x_i).$$

Now, notice that this problem can be rewritten as $d$ independent optimization problems:

$$\underset{\phi_i}{\arg\min} \; - \underset{\mathbf{x}\sim\mathbf{X}}{\mathbb{E}} \log f_{\phi_i}(x_i),$$

which each consists of minimizing the expected marginal negative log-likelihood for one of the $d$ random variables. Given that the NLL is a strictly proper scoring rule (Gneiting & Raftery, 2007), we know that it will be minimized if and only if $f_{\phi_i} = f_{\phi_i^\star}$. Since we assume infinite expressivity for $f_{\phi_i}$, we know that such a solution can be learned and thus, solving Problem (8) will recover the ground truth marginals.

**Part 2 (Copula):** Let us now recall the nature of Problem (7), which assumes that the true marginal parameters $\phi_1^\star,\ldots,\phi_d^\star$ are known:

$$\underset{\phi_c}{\arg\min} \quad - \underset{\mathbf{x}\sim\mathbf{X}}{\mathbb{E}} \log c_{\phi_c}\left(F_{\phi_1^\star}(x_1),\ldots,F_{\phi_d^\star}(x_d)\right).$$

Let $\mathbf{U} = [U_1,\ldots,U_d]$ be the random vector obtained by applying the probability integral transform to the $X_i$, i.e., $U_i = F_{\phi_i^\star}(X_i)$. By construction, the marginal distribution of each $U_i$ will be uniform. Problem (7) can be rewritten as:

$$\underset{\phi_c}{\arg\min} \quad - \underset{\mathbf{u}\sim\mathbf{U}}{\mathbb{E}} \log c_{\phi_c}\left(u_1,\ldots,u_d\right),$$

which corresponds to minimizing the expected joint negative log-likelihood. Again, since the NLL is a strictly proper scoring rule, it will be minimized if and only if $c_{\phi_c} = c_{\phi_c^\star}$. Since we assume infinite expressivity for $c_{\phi_c}$, we know that such a solution can be learned and thus, solving Problem (7) will recover the ground truth copula, which is valid by definition.

Finally, since solving Problem (8) yields the true marginals, and solving Problem (7) based on the solution of Problem (8) recovers the true copula, we have that the combination of both solutions yields $p(x_1,\ldots,x_d)$ (see Eq. (16)) and thus they constitute a valid solution to Problem (4).

$\square$

### B.3 Example of Valid and Invalid Decompositions

To illustrate Props. 1 and 2, we use the TACTiS decoder[2] to fit a two-dimensional hand-crafted distribution. The target distribution is built from a copula which is an equal mixture of two Clayton copulas, with parameters $\theta = 9.75$ and $\theta = -0.99$; and from marginals which are a Gamma distribution with parameter $\alpha = 1.99$ and a Double Weibull distribution with parameter $c = 3$.

As was previously shown in Fig. 3, Fig. 14 shows that the attentional copula and the flows can accurately reproduce the target distribution copula and marginals, when trained using the procedure outlined in Sec. 3.2. This is the expected result according to Prop. 2.

Fig. 15 shows the opposite situation: an attentional copula and flows that together accurately reproduce the target distribution, but individually are very far from the target copula and marginals. This is an example of an invalid copula as described in Prop. 1.

---

[2]We slightly modified the attentional copula architecture by removing the forced $U_{[0,1]}$ distribution from the first variable being sampled. This was to allow the attentional copula to be able to learn any distribution on the unit cube.

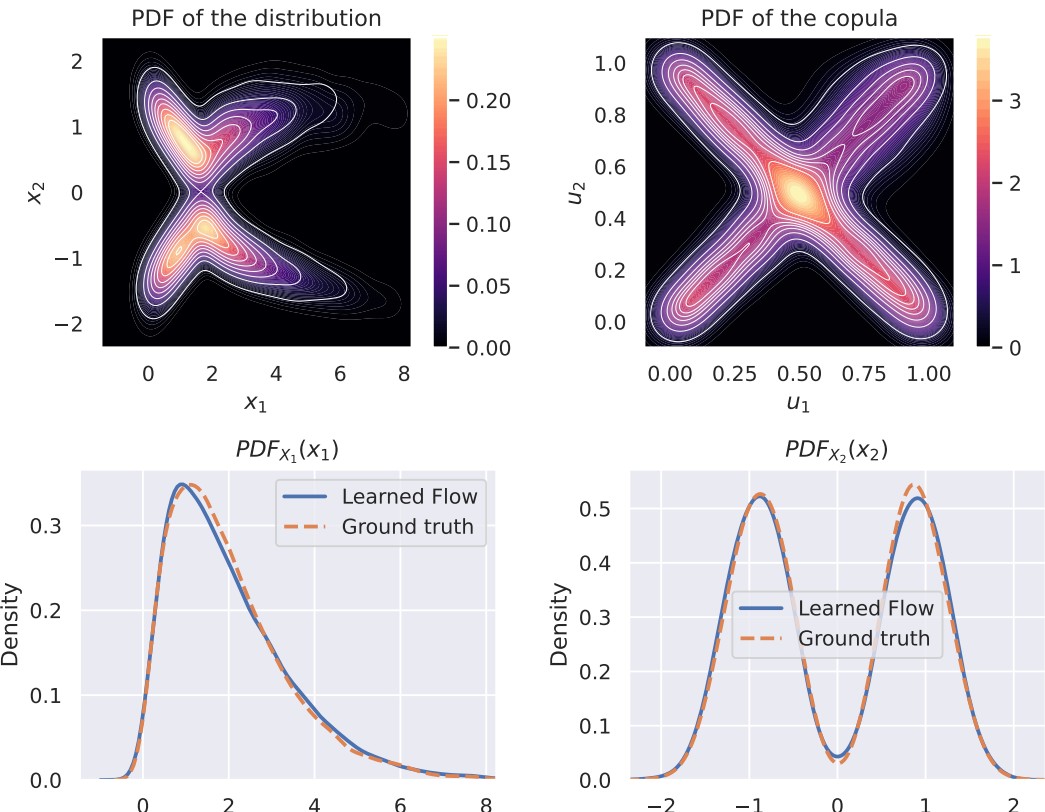

**Figure 14:** *Example of a two-dimensional distribution being accurately split into its constituent copula and marginals using an attentional copula and flows. (top left) Probability Densities of the target distribution (colors) and of the reconstructed distribution (white contours). (top right) Probability Density of the target distribution copula (colors) and of the attentional copula (white contours). (bottom) Probability Densities of each variable marginal distributions (dashed lines) and of the flows (solid lines).*

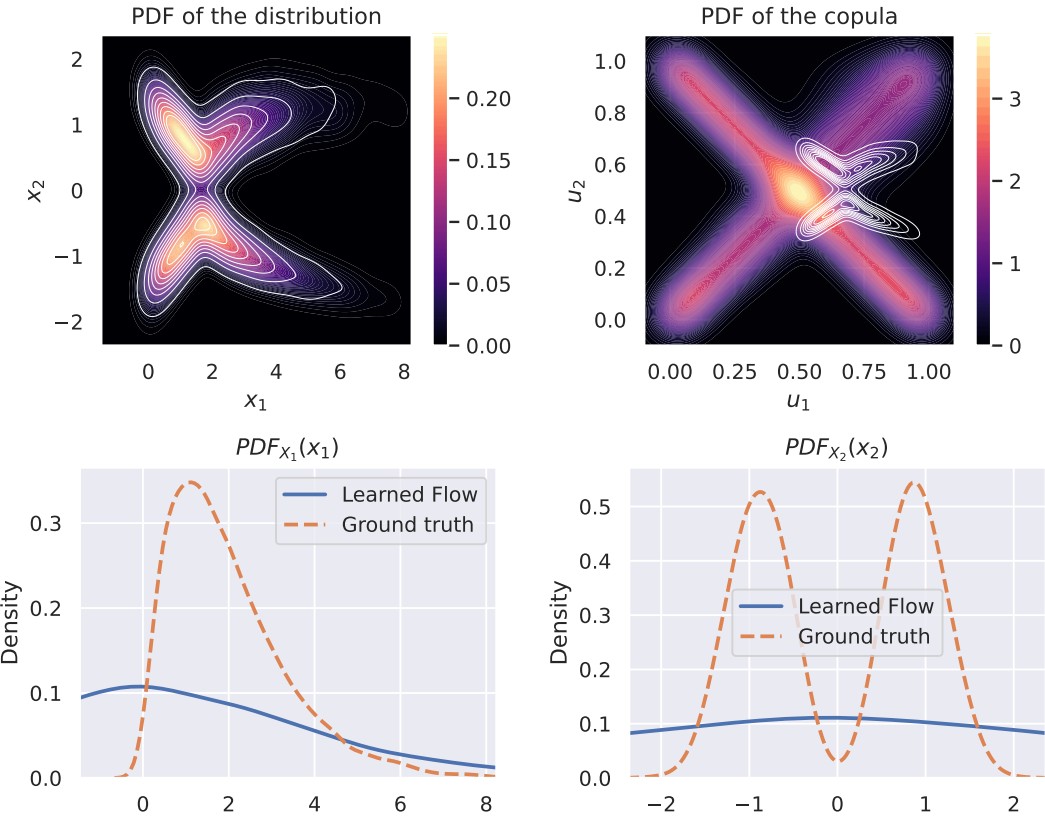

**Figure 15:** *Example of a two-dimensional distribution being inaccurately split into its constituent copula and marginals using an attentional copula and flows. (top left) Probability Densities of the target distribution (colors) and of the reconstructed distribution (white contours). (top right) Probability Density of the target distribution copula (colors) and of the attentional copula (white contours). (bottom) Probability Densities of each variable marginal distributions (dashed lines) and of the flows (solid lines).*

# C   EXPERIMENTAL DETAILS

## C.1   DATASETS

**Table 6:** *Datasets used in the paper. Table reproduced from Drouin et al. (2022) with permission.*

| Short name | Monash name | Frequency | Number of series | Prediction length |
|---|---|---|---|---|
| electricity | Electricity Hourly Dataset | 1 hour | 321 | 24 |
| fred-md | FRED-MD Dataset | 1 month | 107 | 12 |
| kdd-cup | KDD Cup Dataset (without Missing Values) | 1 hour | 270 | 48 |
| solar-10min | Solar Dataset (10 Minutes Observations) | 10 minutes | 137 | 72 |
| traffic | Traffic Hourly Dataset | 1 hour | 862 | 24 |

We reuse the same datasets used by (Drouin et al., 2022) in their forecasting benchmark. Tab. 6 describes the datasets used in the paper for reference.

## C.2   TRAINING PROCEDURE

**Compute used**   All models in the paper are trained in a Docker container with access to a Nvidia Tesla-P100 GPU (12 GB of memory), 2 CPU cores, and 32 GB of RAM. Due to sampling and NLL evaluation requiring more memory due to being done without bagging, they were done with computing resources with more available memory as needed.

**Batch size**   The batch size was selected as the largest power of 2 between 1 and 256 such that the GPU memory requirement from the training loop did not go above the available memory.

**Training loop**   Each epoch of the training loop consists of training the model using 512 batches from the training set, followed by computing the NLL on the validation set. We stop the training when any of these conditions are reached:

- We have reached 72 hours of training for a model without the two-stage curriculum,
- We have reached 36 hours of training for a single stage of the curriculum,
- Or we did not observe any improvement in the best value of the NLL on the validation set for 50 epochs.

We then return the version of the model that reached the best value of the NLL on the validation set.

For SPD, following Biloš (2023), in the absence of NLL, we use the loss function of the model i.e. the squared difference between the predicted and true noise.

**Validation set**   During hyperparameter search, we reserve from the end of the training set a number of timesteps equal to 7 times the prediction length. The validation set is then built from all prediction windows that fit in this reserved data. During backtesting, we also remove this amount of data from the training set (except for fred-md where only remove a number equal to the prediction length), but the validation set is built from the 7 (or single for fred-md) non-overlapping prediction window we can get from this reserved data.

**Interpolation**   For the interpolation experiments, we always use a centered interpolation window: we keep an equal amount of observed timesteps immediately before and immediately after the prediction window. Since we do not have access to the data that happens after the last backtesting prediction window, we instead shift all backtesting prediction windows by the length of the posterior observed data length. All parts of the training set which overlap these backtesting prediction windows are also removed, to prevent any information leakage during training.

## C.3   HYPERPARAMETER SEARCH PROTOCOL

The metric values for Auto-ARIMA, ETS, GPVar, TempFlow, and TimeGrad are taken from Drouin et al. (2022). We refer the reader to Drouin et al. (2022) for details on how the protocol used for hyperparameter

search for these models, but one important detail is that it used CRPS-Sum as the target to minimize. In theory, this should bias the comparison based on CRPS-Sum (Tab. 1) in their favor.

For TACTiS, and TACTiS-2, we use Optuna (Akiba et al., 2019) to minimize the NLL on the validation set. Note that we use the same validation set for the NLL evaluation and the early stopping during hyperparameter search. For each dataset and model, we let Optuna run for 6 days, using 50 training runs in parallel.

For SPD, following Biloš (2023), in the absence of NLL, we used the same protocol as for TACTiS and TACTiS-2 but we minimized the loss function of the model, the squared difference between the predicted and true noise.

For the interpolation experiments, we do not perform a separate hyperparameter search and instead reuse the hyperparameters found for forecasting.

## C.4 SELECTED HYPERPARAMETERS

In this section, we present the possible values for the hyperparameters during the hyperparameter search for TACTiS, TACTiS-2, and SPD, together with their optimal values as found by Optuna. Tabs. 7 to 9 present the hyperparameters which are restricted to finite sets of values for TACTiS, TACTiS-2, and SPD, respectively. Tab. 10 shows the optimal values for the learning rate parameters, which were allowed to be in the $[10^{-6}, 10^{-2}]$ continuous range.

**Table 7:** *Possible hyperparameters for* TACTiS. [e], [f], [k], [s], *and* [t] *respectively indicate the optimal hyperparameters for* `electricity, fred-md, kdd-cup, solar-10min,` *and* `traffic.`

|  | Hyperparameter | Possible values |
|---|---|---|
| Model | Encoder transformer embedding size (per head) and feed-forward network size | 4, 8, 16, 32[et], 64[fs], 128, 256[k], 512 |
|  | Encoder transformer number of heads | 1[k], 2[f], 3[s], 4, 5, 6[t], 7[e] |
|  | Encoder number of transformer layers pairs | 1[k], 2, 3[s], 4, 5[eft], 6, 7 |
|  | Encoder input embedding dimensions | 1[t], 2[e], 3, 4[s], 5, 6, 7[fk] |
|  | Encoder time series embedding dimensions | 5[t], 8, 16[e], 32[k], 48[s], 64, 128, 256[f], 512 |
|  | Decoder DSF number of layers | 1[t], 2, 3[f], 4, 5[es], 6, 7[k] |
|  | Decoder DSF hidden dimensions | 4, 8, 16[fs], 32, 48[e], 64, 128, 256, 512[kt] |
|  | Decoder MLP number of layers | 1[e], 2[ft], 3, 4, 5, 6, 7[ks] |
|  | Decoder MLP hidden dimensions | 4, 8[k], 16, 32[e], 48, 64[f], 128[t], 256, 512[s] |
|  | Decoder transformer number of layers | 1[k], 2[f], 3[e], 4, 5, 6[t], 7[e] |
|  | Decoder transformer embedding size (per head) | 4[s], 8, 16, 32, 48[f], 64[ek], 128[t], 256, 512 |
|  | Decoder number transformer heads | 1[e], 2, 3, 4[ks], 5, 6[ft], 7 |
|  | Decoder number of bins in conditional distribution | 10, 20[ft], 50[s], 100[e], 200[k], 500 |
| Data | Normalization | Standardization[efkst] |
|  | History length to prediction length ratio | 1, 2[ks], 3[eft] |
| Training | Optimizer | Adam[efkst] |
|  | Weight decay | 0[efks], $(10^{-5})$, $(10^{-4})$, $(10^{-3})$[t] |
|  | Gradient clipping | $(10^3)$[efkst], $(10^4)$ |

**Table 8:** *Possible hyperparameters for* TACTiS-2. *$^e$, $^f$, $^k$, $^s$, and $^t$ respectively indicate the optimal hyperparameters for* electricity, fred-md, kdd-cup, solar-10min, *and* traffic.

|  | Hyperparameter | Possible values |
|---|---|---|
| Model | Marginal CDF Encoder transformer embedding size (per head) and feed-forward network size | $4, 8^{st}, 16, 32^k, 64, 128^e, 256, 512^f$ |
|  | Marginal CDF Encoder transformer number of heads | $1, 2, 3^e, 4^{ft}, 5^s, 6^k, 7$ |
|  | Marginal CDF Encoder number of transformer layers pairs | $1^f, 2^s, 3^e, 4^{kt}, 5, 6, 7$ |
|  | Marginal CDF Encoder input encoder layers | $1, 2, 3, 4^{fk}, 5, 6^s, 7^{et}$ |
|  | Marginal CDF Encoder time series embedding dimensions | $5^{fks}, 8^e, 16^t, 32, 48, 64, 128, 256, 512$ |
|  | Attentional Copula Encoder transformer embedding size (per head) and feed-forward network size | $4^k, 8^{fst}, 16^e, 32, 64, 128, 256, 512$ |
|  | Attentional Copula Encoder transformer number of heads | $1, 2^e, 3^f, 4^t, 5^s, 6^k, 7$ |
|  | Attentional Copula Encoder number of transformer layers pairs | $1, 2^s, 3^k, 4^t, 5^{ef}, 6, 7$ |
|  | Attentional Copula Encoder input encoder layers | $1^s, 2^k, 3, 4, 5^t, 6, 7^{ef}$ |
|  | Attentional Copula Encoder time series embedding dimensions | $5, 8^{fk}, 16, 32, 48^s, 64^t, 128, 256^e, 512$ |
|  | Decoder DSF number of layers | $1, 2^s, 3, 4^{kt}, 5, 6^f, 7^e$ |
|  | Decoder DSF hidden dimensions | $4, 8, 16^t, 32, 48^{ef}, 64, 128^k, 256^s, 512$ |
|  | Decoder MLP number of layers | $1, 2^{ef}, 3^{ks}, 4, 5, 6^t, 7$ |
|  | Decoder MLP hidden dimensions | $4, 8, 16^{kst}, 32, 48^f, 64, 128^e, 256, 512$ |
|  | Decoder transformer number of layers | $1, 2^t, 3^s, 4^k, 5, 6, 7^{ef}$ |
|  | Decoder transformer embedding size (per head) | $4, 8, 16^s, 32^e, 48^{kt}, 64^f, 128, 256, 512$ |
|  | Decoder transformer number of heads | $1^k, 2, 3, 4^{tf}, 5, 6^s, 7^e$ |
|  | Decoder number of bins in conditional distribution | $10, 20^e, 50^{fks}, 100^t, 200, 500$ |
| Data | Normalization | Standardization$^{efkst}$ |
|  | History length to prediction length ratio | $1^t, 2^{es}, 3^{fk}$ |
| Training Phase 1 | Optimizer | Adam$^{efkst}$ |
|  | Weight decay | $0^{est}, (10^{-5})^k, (10^{-4}), (10^{-3})^f$ |
|  | Gradient clipping | $(10^3)^{efst}, (10^4)^k$ |
| Training Phase 2 | Optimizer | Adam$^{efkst}$ |
|  | Weight decay | $0^{skt}, (10^{-5}), (10^{-4})^{ef}, (10^{-3})$ |
|  | Gradient clipping | $(10^3)^{es}, (10^4)^{fkt}$ |

**Table 9:** *Possible hyperparameters for SPD. $^e$, $^f$, $^k$, $^s$, and $^t$ respectively indicate the optimal hyperparameters for* electricity, fred-md, kdd-cup, solar-10min, *and* traffic. *The choice of possible hyperparameters was discussed with the authors (Biloš, 2023).*

|  | Hyperparameter | Possible values |
|---|---|---|
| Model | type | `'discrete'`$^{efkst}$, `'continuous'` |
|  | noise | `'normal'`$^{efks}$, `'ou'`, `'gp'`$^t$ |
|  | num_layers | $1, 2^{efs}, 3^{kt}$ |
|  | num_cells | $20^{fkt}, 40^s, 60^e$ |
|  | dropout_rate | $0^k, 0.001^{tf}, 0.01^{es}, 0.1$ |
|  | diff_steps | $25, 50, 100^{efkst}$ |
|  | beta_schedule | `'linear'`, `'quad'`$^{efkst}$ |
|  | residual_layers | $4, 8^e, 16^{fkst}$ |
|  | residual_channels | $4, 8, 16^{efkst}$ |
|  | scaling | `False`$^{ks}$, `True`$^{eft}$ |
| Data | History length to prediction length ratio | $1^{efkst}$ |
| Training | Optimizer | Adam$^{efkst}$ |
|  | Weight decay | $0^s, (10^{-5})^{ek}, (10^{-4})^{ft}, (10^{-3})$ |
|  | Gradient clipping | $(10^1), (10^2), (10^3)^k, (10^4)^{efst}$ |

**Table 10:** *Optimal learning rate as obtained by our hyperparameter search procedure.*

| Model | | electricity | fred-md | kdd-cup | solar-10min | traffic |
|---|---|---|---|---|---|---|
| TACTiS | | $7.4 \times 10^{-5}$ | $2.7 \times 10^{-4}$ | $9.2 \times 10^{-5}$ | $8.0 \times 10^{-5}$ | $2.8 \times 10^{-3}$ |
| TACTiS-2 | Phase 1 | $5.0 \times 10^{-5}$ | $1.7 \times 10^{-3}$ | $2.3 \times 10^{-5}$ | $1.8 \times 10^{-3}$ | $2.5 \times 10^{-3}$ |
| TACTiS-2 | Phase 2 | $2.2 \times 10^{-4}$ | $1.9 \times 10^{-3}$ | $9.4 \times 10^{-4}$ | $7.0 \times 10^{-4}$ | $6.7 \times 10^{-4}$ |
| SPD | | $8.9 \times 10^{-4}$ | $5.6 \times 10^{-3}$ | $2.7 \times 10^{-3}$ | $3.1 \times 10^{-3}$ | $4.4 \times 10^{-3}$ |

# D  ADDITIONAL RESULTS

## D.1  DEMONSTRATION OF MODEL FLEXIBILITY IN REAL-WORLD DATASETS

We present empirical results on real-world datasets verifying the ability of TACTiS-2 to work with uneven and unaligned data. The datasets used in these experiments are derived from real-world datasets with aligned and evenly sampled observations. To create uneven sampling, for each series we independently follow a process where if we keep the observation at timestep $t$, then the next kept observation will be at timestep $t + \delta$, with $\delta$ randomly chosen from 1, 2, or 3. To create unalignment, we randomly choose the sampling frequency for each series from: i) the original frequency, ii) half the original frequency, and iii) a quarter of the original frequency in the dataset. The datasets resulting from such corruptions are faithful to real-world scenarios. We follow the same protocol as described in Sec. 5, except that we use a single backtest timestamp for evaluation. Tab. 11 and Tab. 12 compare the ability of TACTiS and TACTiS-2 to perform forecasting with uneven and unaligned data respectively on the two datasets with the largest forecast horizon. It can be seen that the ability of TACTiS-2 to work with uneven and unaligned data is much more pronounced than that of TACTiS.

**Table 11:** *Mean NLL values for the forecasting experiments on uneven data. Results are over 5 seeds. Lower is better and the best results are in bold.*

| Model | kdd-cup | solar-10min |
|---|---|---|
| TACTiS | $2.858 \pm 0.476$ | $1.001 \pm 0.783$ |
| TACTiS-2 | $\mathbf{1.866 \pm 0.230}$ | $\mathbf{0.261 \pm 0.011}$ |

**Table 12:** *Mean NLL values for the forecasting experiments on unaligned data. Results are over 5 seeds. Lower is better and the best results are in bold.*

| Model | kdd-cup | solar-10min |
|---|---|---|
| TACTiS | $1.527 \pm 0.072$ | $-0.038 \pm 0.220$ |
| TACTiS-2 | $\mathbf{0.722 \pm 0.163}$ | $\mathbf{-3.218 \pm 0.249}$ |

