# OpenReview forum: "TACTiS-2: Better, Faster, Simpler Attentional Copulas for Multivariate Time Series"
_ICLR.cc/2024/Conference — ICLR 2024 poster_

### Official Review · Reviewer_GAmJ · 2023-10-30

**Soundness:** 3 good
**Presentation:** 4 excellent
**Contribution:** 3 good
**Rating:** 8
**Confidence:** 3

**Summary:**

The paper proposes an improvement of TACTiS, which is a permutation-based non-parametric copulas, by adopting transformers and a two-stage training for multi-variate probabilistic time series prediction. This allows to avoid the expensive permutation-based objective, resulting in the number of distributional parameters scaling linearly in the number of variables. The authors numerically show that the resulting model can better train dynamics and achieve state-of-the-art performance, while keeping the flexibility of prior work.

**Strengths:**

•	The author skillfully merges statistical time series analysis with deep learning, estimating the marginal pdfs and attentional copula using autoencoders. This innovative combination results in a more efficient and rapid estimation of probability distributions compared to traditional statistical learning.
•	The proposed model appears highly flexible, accommodating heterogeneous datasets with uneven sampling frequencies. It also demonstrates state-of-the-art accuracies and visually impressive interpolation performance.
•	The author showcases a profound understanding of time series analysis by mathematically defining research problems, propositions, and definitions, accompanied by essential proofs.

**Weaknesses:**

•	While the author elucidates the statistical aspects comprehensively, a more detailed explanation of the autoencoder's use would have been beneficial. For instance, discussing the motivations behind its selection, why it's deemed the best choice, or testing its performance against alternatives like variational autoencoders.
•	In the experiment section, the author evaluates the results using five datasets from the Monash Time Series Forecasting Repository based on dimensions, frequencies, and length. The chosen samples appear to be of a small size. Merely calculating the average rank might not provide an unbiased and comprehensive evaluation. The results would be more persuasive if the author utilized a broader range of datasets and presented a critical difference plot.
•	The author might consider employing other techniques, such as artificially creating uneven sampling frequencies, to garner more samples.

**Questions:**

•	Why using autoencoders rather than other deep learning models such as CNN to estimate the probability distributions?
•	How to capture the dimensional dependency using this architecture?
•	Why do we take this particular subset of datasets?

---

> ### Author Response · Authors · 2023-11-17
>
> Thank you for the very positive assessment of our work and for taking the time to highlight its strengths. Please find a response to your comments and questions below.
>
> **W1 / Q1: Choice of architecture rather than other deep learning models such as CNN to estimate the probability distributions**
>
> The reviewer correctly states that alternative architectures, such as CNNs have been successful in time series tasks. Notable examples are included among our baselines. For instance, GPVar [1] and TimeGrad [2], which use LSTM-based architectures, and SPD [3] which uses a CNN-based architecture. In this work, our choice of using the transformer architecture is based on the added flexibility that it brings to the model [4].
>
> In fact, a key feature of the transformer in TACTiS and TACTiS-2 is how the tokenization is performed for multivariate time series (as described in Sec 2 of our paper). We represent each value in the time series as a token paired with a timestamp and arbitrary stochastic covariates. We define our learning tasks with a mask that indicates which values are observed and which values are to be predicted. Note that this allows any learning task to be defined on the data based on the pattern of the mask. For instance, for forecasting, the values to be predicted would be at the end, while for interpolation, the values to be predicted will be in arbitrary positions in the data, and appropriate masks can be used for these tasks based on the pattern of the observed and predicted values. Arbitrary and more complex tasks can be defined using this approach. Further, as the tokenization does not impose any constraints on the data, this allows the architecture to consume arbitrarily uneven or unaligned data which often occur in the real-world. Thereby, adopting the transformer architecture in TACTiS-2 enables it as a general-purpose model satisfying the five desiderata for real-world time series problems as described in our introduction. We stress that these properties are enabled due to the flexibility of the transformer architecture.
>
> [1] Salinas, D., Bohlke-Schneider, M., Callot, L., Medico, R., and Gasthaus, J. High-dimensional multivariate forecasting with low-rank Gaussian copula processes. Advances in Neural Information Processing Systems, 32: 6827–6837, 2019.
>
> [2] Rasul, K., Seward, C., Schuster, I., & Vollgraf, R. (2021, July). Autoregressive denoising diffusion models for multivariate probabilistic time series forecasting. In International Conference on Machine Learning (pp. 8857-8868). PMLR.
>
> [3] Marin Bilos, Kashif Rasul, Anderson Schneider, Yuriy Nevmyvaka, and Stephan Gunnemann. Modeling temporal data as continuous functions with stochastic process diffusion. In Andreas Krause, Emma Brunskill, Kyunghyun Cho, Barbara Engelhardt, Sivan Sabato, and Jonathan Scarlett (eds.), Proceedings of the 40th International Conference on Machine Learning, volume 202 of Proceedings of Machine Learning Research, pp. 2452–2470. PMLR, 23–29 Jul 2023.
>
> [4] Wen, Q., Zhou, T., Zhang, C., Chen, W., Ma, Z., Yan, J., & Sun, L. (2022). Transformers in time series: A survey. arXiv preprint arXiv:2202.07125.

---

> ### Author Response · Authors · 2023-11-17
>
> **W2 / Q3: Justify the choice of datasets; a broader range of datasets may be utilized**
>
> We take the opportunity to further explain why we used the five datasets - electricity, fred-md, kdd-cup, solar-10min, traffic. These were chosen due to their ubiquity in probabilistic forecasting benchmarks [1, 2, 3, 4]. TimeGrad [1] and TempFlow [2] use Solar, Electricity, Traffic in addition to two others, SPD [3] uses Electricity and Solar, and TACTiS [5] those five.
>
> Nevertheless, we agree that the addition of more datasets will undoubtedly strengthen our contribution. In the short time available, we were able to evaluate the forecasting performance of TACTiS and TACTiS-2 on two new datasets, namely covid-deaths and rideshare, from the Monash Time Series Forecasting Repository [5]. We present these new results below, and add them to Appendix D.1 of the paper. As seen below, the NLL of TACTIS-2 is significantly better than that of TACTIS in both of the added datasets.
>
> Table: Mean NLL values on forecasting. Lower is better.
> | Model    | covid-deaths | rideshare |
> |----------|--------------|-----------|
> | TACTiS   | -1.605 +- 0.121 | 14.815 +- 1.101 |
> | TACTiS-2 | **-4.859 +- 0.093** | **10.451 +- 0.250** |
>
> [1] Rasul, K., Seward, C., Schuster, I., & Vollgraf, R. (2021, July). Autoregressive denoising diffusion models for multivariate probabilistic time series forecasting. In International Conference on Machine Learning (pp. 8857-8868). PMLR.
>
> [2] Rasul, K., Sheikh, A. S., Schuster, I., Bergmann, U. M., & Vollgraf, R. (2020, October). Multivariate Probabilistic Time Series Forecasting via Conditioned Normalizing Flows. In International Conference on Learning Representations.
>
> [3] Marin Bilos, Kashif Rasul, Anderson Schneider, Yuriy Nevmyvaka, and Stephan Gunnemann. Modeling temporal data as continuous functions with stochastic process diffusion. In Andreas Krause, Emma Brunskill, Kyunghyun Cho, Barbara Engelhardt, Sivan Sabato, and Jonathan Scarlett (eds.), Proceedings of the 40th International Conference on Machine Learning, volume 202 of Proceedings of Machine Learning Research, pp. 2452–2470. PMLR, 23–29 Jul 2023.
>
> [4] Alexandre Drouin, Etienne Marcotte, and Nicolas Chapados. TACTiS: Transformer-attentional copulas for time series. In Kamalika Chaudhuri, Stefanie Jegelka, Le Song, Csaba Szepesvari, Gang Niu, and Sivan Sabato (eds.), Proceedings of the 39th International Conference on Machine Learning, volume 162 of Proceedings of Machine Learning Research, pp. 5447–5493. PMLR, 17–23 Jul 2022.
>
> [5] Rakshitha Godahewa, Christoph Bergmeir, Geoffrey I. Webb, Rob J. Hyndman, and Pablo Montero-Manso. Monash time series forecasting archive. In Neural Information Processing Systems Track on Datasets and Benchmarks, 2021.

---

> ### Author Response · Authors · 2023-11-17
>
> **W3: The author might consider employing other techniques, such as artificially creating uneven sampling frequencies, to garner more samples.**
>
> We thank the reviewer for the suggestion. We point to the reviewer that we have added new results to the paper in Appendix D.2 validating the ability of TACTiS-2 to work with uneven and unaligned data, artificially creating uneven and unaligned data from real-world datasets. We agree with the reviewer that it would be interesting to leverage the flexibility of using artificially created samples of unaligned and uneven data as a data augmentation method when training models. Such a method would be orthogonal to TACTiS and can potentially improve its performance.
>
> **Q2: How to capture the dimensional dependency using this architecture?**
>
> One notable property of the TACTiS and TACTiS-2 models is that the multivariate dependencies are guaranteed to be captured by the copula component of the model, not in the others. Hence, if one is interested in inspecting dimensional dependencies to facilitate downstream decision-making tasks, one only needs to focus on the copula. For instance, one could draw samples from the copula and measure the statistical association between all dimensions. This technique was employed to inspect the learned associations in Drouin et al. (2022) [1]. Another interesting fact is that, in this work, we show that TACTiS-2 captures multivariate dependencies much better than TACTiS, as indicated by its significantly lower negative log likelihoods. Hence, this new model is poised to produce more accurate insights into multivariate dependencies, which could in turn lead to more accurate decision-making.
>
> [1] Alexandre Drouin, Etienne Marcotte, and Nicolas Chapados. TACTiS: Transformer-attentional copulas for time series. In Kamalika Chaudhuri, Stefanie Jegelka, Le Song, Csaba Szepesvari, Gang Niu, and Sivan Sabato (eds.), Proceedings of the 39th International Conference on Machine Learning, volume 162 of Proceedings of Machine Learning Research, pp. 5447–5493. PMLR, 17–23 Jul 2022.

---

> > ### Author Response · Authors · 2023-11-21
> >
> > Thank you again for your review. We are wondering if our response was satisfactory or if there are points that need further discussion. We would be happy to engage with you on any of these points before the end of the discussion period.

---

### Official Review · Reviewer_s9Tw · 2023-11-03

**Soundness:** 3 good
**Presentation:** 4 excellent
**Contribution:** 2 fair
**Rating:** 5
**Confidence:** 4

**Summary:**

The paper proposes TACTIS-2, a model for multivariate time series forecasting. TACTIS-2 builds upon TACTIS which uses neural networks to parameterize copulas for multivariate time series forecasting. To ensure the validity of learned copulas, TACTIS trains the model with random permutations of the variables which leads to problems with high-dimensional time series. To address this limitation, TACTIS-2 uses two stage training. In the first stage, marginal distributions are learned without any dependency between them. Later, the copula parameters are learned given the optimal marginal parameters. Forecasting and interpolation results on 5 datasets from Monash time series repository show that TACTIS-2 improves over TACTIS in terms of the prediction performance and training time.

**Strengths:**

- The paper addresses a key limitation in the existing TACTIS model which involves $d!$ factorization of the copula density, in theory. In this work, the authors utilize existing work on copulas to transform the optimization into a two stage problem which scales linearly with the dimensionality.
- The paper is very well written and easy to understand. It discusses the TACTIS model sufficiently for the reader to be able to understand the main contribution.
- The proposed model performs better than TACTIS while being simpler and faster to train.

**Weaknesses:**

- The main weakness of this work is its limited significance. The key contribution is an incremental modification of the existing TACTIS model. While the work may be interesting for individuals specifically focusing on TACTIS, its significance for the broader time series community is unclear. One may argue on the basis of the empirical results; however, in their current form, the results are not exciting and comprehensive enough to fully support this argument (see below).
- In the absence of enough methodological contributions, the empirical contribution needs to be comprehensive. However, the experiments have only been conducted on 5 datasets from the Monash repository. The baseline selection also needs improvement for the _state of the art_ claim. CSDI and SSSD would be better baselines for the empirical comparison.

To improve the paper, consider:
- Adding better baselines such as CSDI and SSSD.
- Comparing on a larger set of datasets from the Monash repository.
- Highlighting other aspects of the model. For instance, the "flexibility to handle unaligned/unevenly-sampled series" is mentioned multiple times in the related work but has only been studied is a toy setting.

**Questions:**

See above.

- Is there a reason why the numbers for electricity and traffic datasets differ so significantly from existing works [1]?

[1] Tashiro, Yusuke, et al. "Csdi: Conditional score-based diffusion models for probabilistic time series imputation." Advances in Neural Information Processing Systems 34 (2021): 24804-24816.

---

> ### Author Response · Authors · 2023-11-17
>
> We thank the reviewer for the constructive feedback and for highlighting the strengths of our work. We address the raised concerns point-by-point and hope that our responses will be satisfactory.
>
> **S1: Addressing the reviewer’s comment that we utilize existing work on copulas**
>
> The reviewer is correct in that we utilize existing work on copulas to transform the optimization into a two-stage problem. However, we re-emphasize that we do not merely apply existing work from the copula literature. We contribute to it by proving that the two-stage approach is still valid in the fully non-parametric setting, where non-parametric estimators are used for the marginal and copula distributions. This result consists of Proposition 2 and its proof in App. B.2. Section 3.2 has been edited to emphasize this point. This result is then used to propose a new optimization problem and architecture to learn attentional copulas (in Sec 4).
>
> **W1: Limited significance, incremental modification of TACTiS, significance to the broader time series community is unclear, absence of enough methodological contributions**
>
> We respectfully disagree. Our work introduces new theoretical results on two-stage optimization of non-parametric copulas, followed by a complete change of the optimization problem in TACTiS (along with necessary architectural changes), and much better predictive performance and training dynamics across a variety of tasks. Our work makes the desirable properties of TACTiS actually usable in practice, leading to what is currently the most accurate general-purpose model for multivariate probabilistic time series prediction. We kindly ask the reviewer to refer to the “Significance of Contribution” section of the general comment for an extended response to their comments and hope that they will reconsider their position.
>
> **W2: Results are not that exciting and comprehensive, empirical contribution needs to be comprehensive**
>
> We have broadened our empirical evaluation following each of the reviewer’s suggestions, as follows.

---

> ### Author Response · Authors · 2023-11-17
>
> **W2-a: Adding CSDI and SSSD as baselines**
>
> We thank the reviewer for the suggestion. We planned to include CSDI [2] as part of our baselines, but our results did not reach the level of performance found in their paper despite our best efforts, and hence we decided to not include it in our forecasting baselines. However, as per the reviewer’s request, we present the results of CSDI and discuss them with respect to those of TACTiS-2 below. We are happy to add this comparison to the final revision of the paper. Regarding SSSD, we agree that it would make a meaningful baseline. It has been left out of our study as the published codebase [4] does not support forecasting out-of-the-box. However, we are happy to revisit this and include it in the final revision of our work.
>
> We hereby present results that compare the CRPS-Sum, CRPS, and Energy Score of CSDI with all other methods on 4 datasets of the forecasting benchmark. We were not able to include the largest dataset (traffic) due to limited time, but we are happy to add it to the final revision of the paper.
>
> Table: Mean CRPS-Sum values for the forecasting experiments. Lower is better.
> | Model      	| electricity 	| fred-md 	| kdd-cup 	| solar-10min 	|
> |-----------------|----------------|-----------|-----------|----------------|
> | Auto-ARIMA 	| 0.077 +- 0.016	| 0.043 +- 0.005	| 0.625 +- 0.066	| 0.994 +- 0.216      	|
> | ETS        	| 0.059 +- 0.011	| 0.037 +- 0.010	| 0.408 +- 0.030 | 0.678 +- 0.097    	|
> | TempFlow   	| 0.075 +- 0.024	| 0.095 +- 0.004 | 0.250 +- 0.010	| 0.507 +- 0.034    	|
> | SPD        	| 0.062 +- 0.016	| 0.048 +- 0.011	| 0.319 +- 0.013	| 0.568 +- 0.061    	|
> | TimeGrad   	| 0.067 +- 0.028  | 0.094 +- 0.030	| 0.326 +- 0.024	| 0.540 +- 0.044    	|
> | GPVar      	| 0.035 +- 0.011  | 0.067 +- 0.008	| 0.290 +- 0.005	| 0.254 +- 0.028   	|
> | CSDI       	| 0.165 +- 0.009 	| 0.043 +- 0.005	| 0.388 +- 0.038	| 1.063 +- 0.081   	|
> | TACTiS     	| 0.021 +- 0.005 	| 0.042 +- 0.009	| 0.237 +- 0.013	| 0.311 +- 0.061	|
> | TACTiS-2   	| **0.020 +-  0.005**	| **0.035 +- 0.005** 	| **0.234 +- 0.011** | **0.240 +- 0.027**   	|
>
> Table: Mean CRPS values for the forecasting experiments. Lower is better.
>
> | Model      	| electricity 	| fred-md 	| kdd-cup 	| solar-10min 	|
> |----------------|-----------------|-----------|-----------|----------------|
> | ETS        	| 0.094 +- 0.014 	| 0.050 +- 0.011 	| 0.560 +- 0.028 	| 0.844 +- 0.119      	|
> | Auto-ARIMA 	| 0.129 +- 0.015	| 0.052 +- 0.005 	| 0.477 +- 0.015 	| 0.636 +- 0.060      	|
> | TempFlow   	| 0.109 +- 0.024 	| 0.110 +- 0.003 	| 0.451 +- 0.005 	| 0.547 +- 0.036     	|
> | TimeGrad   	| 0.101 +- 0.027 	| 0.142 +- 0.058 	| 0.495 +- 0.023 	| 0.560 +- 0.047      	|
> | SPD        	| 0.099 +- 0.016 	| 0.058 +- 0.011 	| 0.465 +- 0.005 	| 0.585 +- 0.050       	|
> | GPVar      	| 0.067 +- 0.010 	| 0.086 +- 0.009	| 0.459 +- 0.009 	| 0.298 +- 0.034       	|
> | CSDI       	| 0.182 +- 0.011 	| 0.055 +- 0.006 	| 0.527 +- 0.028 	| 1.053 +- 0.067      	|
> | TACTiS     	| 0.052 +- 0.006 	| 0.048 +- 0.010 	| 0.420 +- 0.007 	| 0.326 +- 0.049      	|
> | TACTiS-2   	| **0.049 +- 0.006** 	| **0.043 +- 0.006**	| **0.413 +- 0.007** 	| **0.256 +- 0.029**       	|
>
> Table: Mean Energy Score values for the forecasting experiments. Lower is better.
> | Model      | electricity (x 10000) | fred-md (x 100000) | kdd-cup (x 1000) | solar-10min (x 100) |
> |------------|-----------------------|--------------------|-----------------|---------------------|
> | Auto-ARIMA | 44.59 +- 8.56         | 8.72  +- 0.81      | 18.76  +- 3.31  | 19.42 +- 3.37      |
> | TempFlow   | 10.25 +- 2.03         | 20.16 +- 0.74      | 3.30   +- 0.19  | 4.25 +- 0.16         |
> | ETS        | 7.94  +- 0.93         | 7.90  +- 1.88      | 3.60   +- 0.24  | 4.74  +- 0.17      |
> | TimeGrad   | 9.69  +- 2.62         | 19.87  +- 7.23     | 3.30   +- 0.19  | 4.31 +- 0.23         |
> | SPD        | 8.90  +- 1.18         | 8.94  +- 1.82      | 3.13   +- 0.24  | 3.68  +- 0.31      |
> | GPVar      | 6.80  +- 0.62         | 11.43 +- 1.60      | 3.18  + 0.20    | 2.60 +- 0.10         |
> | CSDI       | 19.96 +- 0.93         | 7.97  +- 0.01      | 3.37 +- 0.21    | 7.74 +- 0.66         |
> | TACTiS     | 5.42  +- 0.57         | 8.18  +- 1.83      | 2.93 +- 0.22    | 2.88 +- 0.23         |
> | TACTiS-2   | **4.91  +- 0.52**         | **6.72  +- 0.10**      | **2.81  +- 0.19**  | **2.37 +- 0.14**         |

---

> > ### Comment · Reviewer_s9Tw · 2023-11-22
> >
> > Thank you for posting these results. I have some concerns regarding this evaluation. I have worked with the CSDI model in the past and CSDI being significantly worse than all these baselines is difficult to believe. Could there be an issue with the training and evaluation of CSDI?

---

> ### Author Response · Authors · 2023-11-17
>
> As the three tables above show, CSDI is far from all the other baselines in the electricity and solar-10min datasets, whereas in the fred-md and kdd-cup datasets, TACTiS-2 surpasses it by a large amount.
>
> Finally, we would like to point out that the baseline SPD [1], which we compare with, is more recent than both CSDI [2] and SSSD [3], and that it is the only one, apart from the TACTiS models, that satisfies all of the five desiderata for a general-purpose model mentioned in the introduction. Our claim that TACTiS-2 is state-of-the-art is supported by its significant improvements over a variety of classical methods, transformer-based approaches, copula-based ones, and the recent and flexible SPD. We are nevertheless happy to add CSDI and SSSD in the final revision of the paper.
>
> [1] Marin Bilos, Kashif Rasul, Anderson Schneider, Yuriy Nevmyvaka, and Stephan Gunnemann. Modeling temporal data as continuous functions with stochastic process diffusion. In Andreas Krause, Emma Brunskill, Kyunghyun Cho, Barbara Engelhardt, Sivan Sabato, and Jonathan Scarlett (eds.), Proceedings of the 40th International Conference on Machine Learning, volume 202 of Proceedings of Machine Learning Research, pp. 2452–2470. PMLR, 23–29 Jul 2023.
>
> [2] Yusuke Tashiro, Jiaming Song, Yang Song, and Stefano Ermon. CSDI: Conditional score-based diffusion models for probabilistic time series imputation. In Advances in Neural Information Processing Systems, volume 34, 2021.
>
> [3] Juan Lopez Alcaraz and Nils Strodthoff. Diffusion-based time series imputation and forecasting with structured state space models. Transactions on Machine Learning Research, 2023. ISSN 2835-8856.
>
> [4] AI4HealthUOL, Codebase of Diffusion-based Time Series Imputation and Forecasting with Structured State Space Models (SSSD), 2023, GitHub repository, https://github.com/AI4HealthUOL/SSSD

---

> > ### Comment · Reviewer_s9Tw · 2023-11-22
> >
> > Based on a quick look at the SSSD repo, they seem to have configs for missing value setups but it should not be difficult to use it for forecasting. This would definitely improve the quality of the experiments. Also, check out the conditional model proposed in [1] which is similar to SSSD.
> >
> > [1] Kollovieh, Marcel, et al. "Predict, refine, synthesize: Self-guiding diffusion models for probabilistic time series forecasting." arXiv preprint arXiv:2307.11494 (2023).

---

> > > ### Author Response · Authors · 2023-11-22
> > >
> > > **Thank you for your response**
> > >
> > > We are glad to hear that you do not see any blocker to the acceptance of this work.

---

> > > > ### Author Response · Authors · 2023-11-22
> > > >
> > > > **Regarding the discrepancy in the performance of CSDI**:
> > > >
> > > > We were also surprised by this discrepancy. To try to understand and fix it, we were in contact with the corresponding author of CSDI, sharing code, protocol, and hyperparameters, and were still not able to resolve it. We ensure that the protocol used for this experiment was as rigorous as the rest of the experimental work in the paper and will make all code available upon acceptance to allow reproduction.

---

> > > > > ### Author Response · Authors · 2023-11-22
> > > > >
> > > > > **Regarding SSSD**: We have extensively looked at this GitHub repository and agree with your intuition. However, as reflected by questions from users in the repository’s issues, this is not as straightforward as it seems given the current code base. Nonetheless, we agree that this is an interesting baseline and will include it in the next revision. We however emphasize that this baseline is not a general-purpose method, since it does not encompass unaligned and unevenly-sampled time series. Our current results rightly support our claims for significant improvement over existing general-purpose methods (TACTiS [1] and SPD [2]).
> > > > >
> > > > > **Regarding Kollovieh et al.**: Thank you for sharing this interesting concurrent work. We will consider it as a baseline for future work.
> > > > >
> > > > > [1] Alexandre Drouin, Etienne Marcotte, and Nicolas Chapados. TACTiS: Transformer-attentional copulas for time series. In Kamalika Chaudhuri, Stefanie Jegelka, Le Song, Csaba Szepesvari, Gang Niu, and Sivan Sabato (eds.), Proceedings of the 39th International Conference on Machine Learning, volume 162 of Proceedings of Machine Learning Research, pp. 5447–5493. PMLR, 17–23 Jul 2022.
> > > > >
> > > > > [2] Marin Bilos, Kashif Rasul, Anderson Schneider, Yuriy Nevmyvaka, and Stephan Gunnemann. Modeling temporal data as continuous functions with stochastic process diffusion. In Andreas Krause, Emma Brunskill, Kyunghyun Cho, Barbara Engelhardt, Sivan Sabato, and Jonathan Scarlett (eds.), Proceedings of the 40th International Conference on Machine Learning, volume 202 of Proceedings of Machine Learning Research, pp. 2452–2470. PMLR, 23–29 Jul 2023.

---

> ### Author Response · Authors · 2023-11-17
>
> **W2-b: Comparing on a larger set of datasets from the Monash repository.**
>
> Following the reviewer’s suggestion, we conduct experiments on two additional datasets from the Monash repository. Preliminary results are reported below, which reinforce that the performance of TACTiS-2 is superior to that of TACTiS.
>
> However, let us start by motivating the choice of five datasets considered in the paper: electricity, fred-md, kdd-cup, solar-10min, traffic. These were chosen due to their ubiquity in probabilistic forecasting benchmarks [1, 2, 3, 4]. TimeGrad [1] and TempFlow [2] use Solar, Electricity, Traffic in addition to two others, SPD [3] uses Electricity and Solar, and TACTiS [5] those five. We further point out that these datasets cover a wide range of dimensionality (n ∈ [107, 826]), sampling frequencies (monthly, hourly, and 10 min.), and prediction lengths (ℓ ∈ [12, 72]) that contributes to a comprehensive evaluation in our work.
>
> Nevertheless, we agree that the addition of more datasets will undoubtedly strengthen our contribution. In the short time available, we were able to evaluate the forecasting performance of TACTiS and TACTiS-2 on two new datasets, namely covid-deaths and rideshare, from the Monash Time Series Forecasting Repository [5]. We present these new results below, and add them to Appendix D.1 of the paper. As seen below, the NLL of TACTIS-2 is significantly better than that of TACTIS in both of the added datasets.
>
> Table: Mean NLL values on forecasting. Lower is better.
> | Model    | covid-deaths | rideshare |
> |----------|--------------|-----------|
> | TACTiS   | -1.605 +- 0.121 | 14.815 +- 1.101 |
> | TACTiS-2 | **-4.859 +- 0.093** | **10.451 +- 0.250** |
>
> [1] Rasul, K., Seward, C., Schuster, I., & Vollgraf, R. (2021, July). Autoregressive denoising diffusion models for multivariate probabilistic time series forecasting. In International Conference on Machine Learning (pp. 8857-8868). PMLR.
>
> [2] Rasul, K., Sheikh, A. S., Schuster, I., Bergmann, U. M., & Vollgraf, R. (2020, October). Multivariate Probabilistic Time Series Forecasting via Conditioned Normalizing Flows. In International Conference on Learning Representations.
>
> [3] Marin Bilos, Kashif Rasul, Anderson Schneider, Yuriy Nevmyvaka, and Stephan Gunnemann. Modeling temporal data as continuous functions with stochastic process diffusion. In Andreas Krause, Emma Brunskill, Kyunghyun Cho, Barbara Engelhardt, Sivan Sabato, and Jonathan Scarlett (eds.), Proceedings of the 40th International Conference on Machine Learning, volume 202 of Proceedings of Machine Learning Research, pp. 2452–2470. PMLR, 23–29 Jul 2023.
>
> [4] Alexandre Drouin, Etienne Marcotte, and Nicolas Chapados. TACTiS: Transformer-attentional copulas ´ for time series. In Kamalika Chaudhuri, Stefanie Jegelka, Le Song, Csaba Szepesvari, Gang Niu, and Sivan Sabato (eds.), Proceedings of the 39th International Conference on Machine Learning, volume 162 of Proceedings of Machine Learning Research, pp. 5447–5493. PMLR, 17–23 Jul 2022.
>
> [5] Rakshitha Godahewa, Christoph Bergmeir, Geoffrey I. Webb, Rob J. Hyndman, and Pablo Montero-Manso. Monash time series forecasting archive. In Neural Information Processing Systems Track on Datasets and Benchmarks, 2021.

---

> ### Author Response · Authors · 2023-11-17
>
> **W2-c: Highlight other aspects of the model. For instance, flexibility to handle unaligned/uneven data is only done on toy datasets.**
>
> We thank the reviewer for suggesting the use of real-world data to evaluate the flexibility properties of TACTiS-2, as it rightfully improves the experimental quality of the work. We here report results on forecasting for real-world datasets where the observations have been rendered unevenly spaced and unaligned.
>
> **Protocol and Results**: The datasets used in these experiments have been derived from real-world datasets with aligned and evenly sampled observations. To create uneven sampling, we randomly skipped the observations at some time steps in the original data. To create unalignment, we randomly selected a sampling frequency for each series in the dataset. The datasets resulting from such corruptions are faithful to real-world scenarios. Due to time constraints, we perform experiments on the kdd-cup and solar-10min datasets. These were chosen since they have the longest prediction horizon (48 and 72, respectively). A detailed discussion of the setting and implementational details has been added to the Appendix in Section D.2, and we will add results on the other datasets in the final version of the paper. As shown below, TACTiS-2 outperforms TACTiS in forecasting in real-world datasets where the data is uneven or unaligned.
>
> Table: Mean NLL values of forecasting on uneven data. Lower is better.
> | Model    | kdd-cup     | solar-10min   |
> |----------|---------|---------|
> | TACTiS   | 2.858 +- 0.476| 1.001 +- 0.783 |
> | TACTiS-2 | **1.866 +- 0.230**| **0.261 +- 0.011** |
>
> Table: Mean NLL values of forecasting on unaligned data. Lower is better.
> | Model    | kdd-cup     | solar-10min   |
> |----------|---------|---------|
> | TACTiS   | 1.527 +- 0.072 | -0.038 +- 0.220 |
> | TACTiS-2 | **0.722 +- 0.163** | **-3.218 +- 0.249** |
>
> **Q1: Why do numbers for electricity and traffic differ from existing work like in CSDI?**
>
> In CSDI [1] the training protocol consists in using all data prior to a fixed date for training and testing on rolling windows over the subsequent data, adopting the protocol from GPVar [2]. Our work instead uses the backtesting protocol of TACTiS [3], which combines rolling-window evaluation with periodic retraining on a series of timestamps, averaging the results across these timestamps. Such a procedure mimics the use of the model in a real-world setting.
>
> [1] Yusuke Tashiro, Jiaming Song, Yang Song, and Stefano Ermon. CSDI: Conditional score-based diffusion models for probabilistic time series imputation. In Advances in Neural Information Processing Systems, volume 34, 2021.
>
> [2] Salinas, D., Bohlke-Schneider, M., Callot, L., Medico, R., and Gasthaus, J. High-dimensional multivariate forecasting with low-rank Gaussian copula processes. Advances in Neural Information Processing Systems, 32: 6827–6837, 2019.
>
> [3] Alexandre Drouin, Etienne Marcotte, and Nicolas Chapados. TACTiS: Transformer-attentional copulas ´ for time series. In Kamalika Chaudhuri, Stefanie Jegelka, Le Song, Csaba Szepesvari, Gang Niu, and Sivan Sabato (eds.), Proceedings of the 39th International Conference on Machine Learning, volume 162 of Proceedings of Machine Learning Research, pp. 5447–5493. PMLR, 17–23 Jul 2022.

---

> > ### Author Response · Authors · 2023-11-21
> >
> > Thank you again for your review. We are wondering if our response was satisfactory or if there are points that need further discussion in order for you to reconsider your current score. We would be happy to engage with you on any of these points before the end of the discussion period.

---

> > ### Comment · Reviewer_s9Tw · 2023-11-22
> >
> > Thank you for your comprehensive response and for the new results. Overall, my opinion of the paper has improved and I won't oppose the acceptance of this paper. However, my main concerns on the significance and the empirical evaluation still exist to some degree. Hence, I cannot recommend acceptance confidently.

---

### Official Review · Reviewer_sGoi · 2023-11-04

**Soundness:** 3 good
**Presentation:** 3 good
**Contribution:** 3 good
**Rating:** 6
**Confidence:** 3

**Summary:**

This paper introduces a new model for multivariate probabilistic time series prediction. Specifically, it improves the (parameter and computation) efficiency of the previous method TACTiS by introducing a simplified objective and the corresponding learning algorithms and neural network architectures. Experiments show that the proposed method can achieve state-of-the-art performance with less computation.

**Strengths:**

- The proposed technique is well-motivated. It identifies the drawbacks and the reasons for the previous method TACTiS and designs specific methods to address that.
- The paper is well-written.

**Weaknesses:**

- Although TACTiS-2 is generally well motivated, it is still unclear how the two-stage solution in Sec 3.2 is derived. It should be clarified whether it is derived based on any assumptions/theorem or directly constructed.
- Can the two-stage optimization achieve the optimal solution of the optimization problem in Eq. (7) and (8)? Although the optimal solution of each sub-problem in Eq (7) and (8) can be achieved, the gap between the theoretical optimal solution and the two-stage method should be discussed clearly.
- The gap between TACTiS and TACTiS-2 can be further discussed.
- May the author explain more about the difference between the task “solar-10min” and others, considerin the different behavior on it, as shown in Figure 1.

**Questions:**

Please clarify the questions mentioned in "Weaknesses".

---

> ### Author Response · Authors · 2023-11-17
>
> We thank the reviewer for the positive evaluation of our work. We are glad to clarify the points brought up in the review.
>
> **W1: Unclear how the two-stage solution in Sec 3.2 is derived. Any assumptions/theorem or is directly constructed.**
>
> Our proposed approach to learning non-parametric copulas is theoretically supported by Proposition 2 (Validity), which shows that any non-parametric copula learned using the proposed two-stage approach is valid. As stated in Sec. 3.2, the proof of this proposition is in App B.2. This proof relies heavily on Sklar's theorem [1]. It assumes that the random variables have continuous marginals and that the marginal and copula estimators have infinite expressivity (capacity in practical terms).
>
> [1] Abe Sklar. Fonctions de répartition à n dimensions et leurs marges. Publications de l’Institut Statistique de l’Université de Paris, 8:229–231, 1959.
>
> **W2: Can the two-stage optimization achieve the optimal solution of (7) and (8)? Although the optimal solution of each sub-problem in Eq (7) and (8) can be achieved, the gap between the theoretical optimal solution and the two-stage method should be discussed clearly.**
>
> Yes, Proposition 2 shows that, given estimators with sufficient capacity and continuous random variables, the two-stage procedure is guaranteed to attain the optimum for both Problems (7) and (8) and that this also corresponds to the minimum of Problem (4). We refer the reviewer to App. B.2 for a detailed proof.
>
> Beyond the theoretical guarantee provided by Proposition 2, Section 5.1 brings empirical evidence that the optimal solution can even be attained in conditions of finite amounts of data, model capacity, and training time. Using a distribution where the ground-truth copula is known, Fig 3 shows that our two-stage approach leads to a learned copula density that closely matches the ground truth. Fig 14 in the Appendix Section B.3 further shows that the learned marginal distributions closely match the ground truth. Hence, this experiment confirms that our two-stage procedure can reach the optimal solution of Problems (7) and (8), respectively, in a practical setting.
>
> As for real-world datasets where the ground-truth copula is not known, we remark that it is not possible to verify that a solution is optimal. Yet, we now provide evidence that the two-stage procedure reaches solutions that are closer to the optimum (i.e., the minimum of Problem (4)) than alternative approaches:
>
> 1. Using the permutation-based approach of Drouin et al. (2022) [1] (denoted as TACTiS), and
>
> 2. Optimizing Problem (4) directly without the two-stage procedure (denoted as TACTiS-2 without the curriculum).
>
> This can be observed in the following table, which shows that, out of all considered approaches, the two-stage curriculum used in TACTiS-2 leads to the smallest negative log-likelihoods, indicating that TACTiS-2's solutions are the closest to optimality.
>
> Table: Mean NLL values on forecasting. Results are over a single backtesting timestamp. Lower is better.
> | Model                           | KDD      | Solar  |
> |-----------------------------|----------|--------|
> | TACTiS                          | 0.489 +- 0.095 | -1.458 +- 0.278 |
> | TACTiS-2 without the curriculum | -0.086 +- 0.206| -2.654 +- 0.192 |
> | TACTiS-2                        | **-1.343 +- 0.055**| **-5.107 +- 0.287** |

---

> ### Author Response · Authors · 2023-11-17
>
> **W3: The gap between TACTiS and TACTiS-2 can be further discussed.**
>
> In addressing this comment, we interpret “the gap” as “the empirical performance gap”. If it was meant as “methodological gap”, we invite the reviewer to refer to the “significance of contribution” paragraph in the general comment for a detailed exposition of the differences between TACTiS and TACTiS-2. We will be glad to further discuss any other interpretation.
>
> TACTiS-2’s superior performance is mainly due to its improved optimization procedure, which ultimately allows it to reach better solutions, in particular better values for the copula parameters. In contrast, the convergence of TACTiS is significantly slower (see Fig. 4). This likely results from: i) optimizing a much more complex objective, with a factorial parameter complexity (see Sec. 3.1), ii) jointly optimizing the marginal and copula components, and iii) from the difficulty of achieving permutation invariance, which is essential to its convergence and for the validity of the copula. Additionally, the performance of TACTiS-2 is further enhanced by its use of a dual-encoder that learns representations specialized for each distributional component.
>
> We added this valuable discussion on the empirical gap in the revised Section 7.
>
> [1] Alexandre Drouin, Etienne Marcotte, and Nicolas Chapados. TACTiS: Transformer-attentional copulas for time series. In Kamalika Chaudhuri, Stefanie Jegelka, Le Song, Csaba Szepesvari, Gang Niu, and Sivan Sabato (eds.), Proceedings of the 39th International Conference on Machine Learning, volume 162 of Proceedings of Machine Learning Research, pp. 5447–5493. PMLR, 17–23 Jul 2022.
>
> **W4: The difference between solar-10min and others:**
>
> We thank the reviewer for the opportunity to discuss this difference. The solar-10min dataset taken from Lai et al. [1], contains solar power production records sampled every 10 minutes from 137 PV plants in Alabama. Among the considered datasets (see Appendix C.1 Table 6), this is the one with the highest sampling frequency (10 minutes). Therefore, such a dataset is rich in statistical dependencies due to the observations being very close in time. In addition, the geographical proximity of the PV plants leads to non-negligible correlations between the series due to shared factors, such as weather conditions.
>
> Indeed, the largest improvement in NLL is observed on the solar-10min dataset, as seen in Figure 1. We explain this observation by TACTiS-2’s improved ability to model multivariate dependencies. As for the drop in FLOPs, as shown in Table 2, TACTiS-2 still reduces the FLOPs by a significant amount. However the drop in FLOPs in less compared to other datasets, possibly because TACTiS was already quite efficient in FLOPs for this dataset, leaving less room for improvements by TACTiS-2.
>
> [1] Lai, Guokun, et al. "Modeling long-and short-term temporal patterns with deep neural networks." The 41st international ACM SIGIR conference on research & development in information retrieval. 2018.

---

> > ### Author Response · Authors · 2023-11-21
> >
> > Thank you again for your review. We are wondering if our response was satisfactory or if there are points that need further discussion. We would be happy to engage with you on any of these points before the end of the discussion period.

---

### Official Review · Reviewer_NV8k · 2023-11-07

**Soundness:** 3 good
**Presentation:** 3 good
**Contribution:** 3 good
**Rating:** 5
**Confidence:** 3

**Summary:**

The paper presents an advanced model for multivariate time series prediction that excels in forecasting and interpolation tasks. By applying copula theory, the authors propose a scalable transformer-based model with linear parameterization growth and a new training curriculum. This model outperforms existing benchmarks in real-world forecasting while adeptly managing irregularly sampled data. The paper details a modification of an existing approach named TACTiS, reducing computational complexity with a linear-scaling parameterization and a new training curriculum, enhancing performance on forecasting tasks and handling irregular data.

**Strengths:**

- The paper addresses the challenging problem of estimating joint predictive distributions for high-dimensional time series data, which has broad applicability across numerous fields.
- It introduces a universal model framework that transcends the need for domain-specific models, potentially streamlining predictive analysis in various applications.
- The Two-stage curriculum approach simplifies the optimization process, which is beneficial for practical implementations.

**Weaknesses:**

- The core innovation claimed by the paper is the reduction in computational complexity through a two-stage solution, first estimating marginals and then dependencies. However, this approach isn't novel, as seen in references [1,2]. The paper would benefit from a clearer distinction of how its methodology differs significantly from these existing methods.
- The paper's primary contribution seems to be an incremental advancement in efficiency over the TACTiS approach. More substantial evidence or arguments are needed to establish this as a significant contribution to the field.
- When evaluating the model's efficacy, the improvement in terms of Negative Log-Likelihood (NLL) is notable. However, the Mean Continuous Ranked Probability Score (CRPS) metric indicates that these improvements are only marginal when compared to the TACTiS model.

[1] Andersen, Elisabeth Wreford. "Two-stage estimation in copula models used in family studies." Lifetime Data Analysis 11 (2005)

[2] Joe, Harry. "Asymptotic efficiency of the two-stage estimation method for copula-based models." Journal of Multivariate Analysis 94.2 (2005).

**Questions:**

- The core innovation claimed by the paper is the reduction in computational complexity through a two-stage solution, first estimating marginals and then dependencies. However, this approach isn't novel, as seen in references [1,2]. The paper would benefit from a clearer distinction of how its methodology differs significantly from these existing methods.
- The paper's primary contribution seems to be an incremental advancement in efficiency over the TACTiS approach. More substantial evidence or arguments are needed to establish this as a significant contribution to the field.
- When evaluating the model's efficacy, the improvement in terms of Negative Log-Likelihood (NLL) is notable. However, the Mean Continuous Ranked Probability Score (CRPS) metric indicates that these improvements are only marginal when compared to the TACTiS model.

[1] Andersen, Elisabeth Wreford. "Two-stage estimation in copula models used in family studies." Lifetime Data Analysis 11 (2005)

[2] Joe, Harry. "Asymptotic efficiency of the two-stage estimation method for copula-based models." Journal of Multivariate Analysis 94.2 (2005).

---

> ### Author Response · Authors · 2023-11-17
>
> We thank the reviewer for the thorough review of our work and for taking the time to acknowledge its strengths. In the following response, we carefully address the raised concerns. We hope that our answers are satisfactory and that the original rating will be reconsidered.
>
> **W1: The core innovation is reduction in computational complexity through a two-stage solution. However this approach isn’t novel as seen in references [1] and [2].**
>
> The reviewer is correct that such a two-stage procedure has previously been studied in the copula literature. In fact, we had attributed this approach to Joe and Xu (1996) [3] in the paper and have now added the relevant references that you suggested in the revision of the paper. As for the novelty component, we would like to emphasize that, to the best of our knowledge, previous work [1, 2, 3] has shown the validity of this approach in the context of parametric and semi-parametric copula estimation. In contrast, the use of neural networks for both the copula and marginals places us in the fully non-parametric setting. For this reason, we contribute Proposition 2 (and its proof in App. B.2), which shows that this approach is still valid in the non-parametric setting. We then proceed to using this result to make significant improvements over known approaches to learning transformer-based non-parametric copulas, constituting an additional element of novelty. We refer the reviewer to the “Significance of Contribution” in the General Comment for a detailed discussion of the significance of our work.
>
> [1] Andersen, Elisabeth Wreford. “Two-stage estimation in copula models used in family studies.” Lifetime Data Analysis 11 (2005)
>
> [2] Joe, Harry. “Asymptotic efficiency of the two-stage estimation method for copula-based models.” Journal of Multivariate Analysis 94.2 (2005).
>
> [3] Joe, Harry, and James Jianmeng Xu. 1996. “The Estimation Method of Inference Functions for Margins for Multivariate Models.” R. Faculty Research and Publications. October 31.
>
> **W2: Incremental advancement in efficiency over TACTIS. More substantial evidence or arguments are needed to establish this as a significant contribution to the field.**
>
> The reviewer is correct that our contributions result in an improvement in efficiency in comparison with TACTiS. However, the main outcome of this work is not merely a minor speedup in model training. Theoretically, the reduced parameter complexity (from factorial to linear) is a strong result in itself. Empirically, Figure 4 and App. A.3 show that TACTiS-2 converges using a fraction of the compute required by TACTiS to converge over a period of **three days**. This fact has significant practical implications (much less wasted compute, enabling more frequent retrainings, among others). Further, in addition to being more efficient, TACTiS-2 converges to much better solutions. This is reflected by improved test metrics across all tasks, which have been acknowledged by all reviewers. In conclusion, our work addresses the drawbacks (sGoi) and key limitations (s9Tw) of TACTiS, making significant progress toward unlocking the full potential of such architectures for such general-purpose modeling of time series distributions. Rightly as the reviewer points out in the strengths, such a contribution is bound to have a broad applicability across numerous fields.

---

> > ### Author Response · Authors · 2023-11-21
> >
> > Thank you again for your review. We are wondering if our response was satisfactory or if there are points that need further discussion in order for you to reconsider your current score. We would be happy to engage with you on any of these points before the end of the discussion period.

---

> ### Author Response · Authors · 2023-11-17
>
> **W3: The improvement in NLL is notable, but the mean CRPS indicates that the improvement is only marginal when compared to TACTIS.**
>
> We agree with the reviewer that TACTiS-2 is only marginally better than TACTiS in terms of CRPS(-Sum), while still largely outperforming other baselines. This is not surprising given that both models share the same architecture for modeling marginal distributions. In fact, by definition, the CRPS only measures errors in marginal distributions. The CRPS-Sum was introduced as an alternative that accounts for multivariate dependencies and has since been established as a community standard. However, recent work emphasized that this metric is more sensitive to errors in marginal distributions than multivariate dependencies [1, 2]. Our contributions mostly affect the learning of multivariate dependencies (copula), which explains the reviewer’s observation.
>
> The NLL is known to be much more sensitive to errors in multivariate dependencies [2] and thus, we use it to further compare the models. As expected, TACTiS-2 achieves much smaller NLLs, indicating that it significantly outperforms TACTiS in learning the copula.
>
> [1] Koochali, A., Schichtel, P., Dengel, A., and Ahmed, S. Random noise vs. state-of-the-art probabilistic forecasting methods: A case study on CRPS-Sum discrimination ability. Applied Sciences, 12(10) :5104, 2022.
>
> [2] Étienne Marcotte, Valentina Zantedeschi, Alexandre Drouin, and Nicolas Chapados. Regions of reliability in the evaluation of multivariate probabilistic forecasts. In Andreas Krause, Emma Brunskill, Kyunghyun Cho, Barbara Engelhardt, Sivan Sabato, and Jonathan Scarlett (eds.), Proceedings of the 40th International Conference on Machine Learning, volume 202 of Proceedings of Machine Learning Research, pp. 23958–24004. PMLR, 23–29 Jul 2023.

---

> ### Comment · Reviewer_NV8k · 2023-11-22
> **Response to authors R#NV8k. (1/2)**
>
> Thanks to the authors for their efforts in preparing the response.
>
> **Novelty Concern**:
> Concerning the novel aspects of your study, my understanding is that its primary innovation lies in the application of a two-stage procedure within non-parametric settings. To enhance the clarity and impact of your contribution, it would be beneficial to explicitly detail the specific modifications you made to this procedure. Transitioning it from a parametric to a non-parametric context is advantageous, but it might be viewed as a relatively incremental advancement. Highlighting your distinct changes would better underscore the novelty of your approach.
>
> **effiency contribution**: "The reduced parameter complexity (from factorial to linear) is a strong result in itself." I agree with the authors, however, I would like to clarify whether this improvement is primarily attributed to the two-stage approach, which was pre-existing?

---

> ### Comment · Reviewer_NV8k · 2023-11-22
> **Response to authors R#NV8k. (2/2)**
>
> Thanks to the authors for your comprehensive response, which I found clear and convincing. However, there remains one aspect I'd like to understand better. You mentioned, 'This is not surprising given that both models share the same architecture for modeling marginal distributions.' Could you clarify how the models, despite having identical architecture, differ in their performance, especially in terms of CRPS (-sum)? This clarification would be greatly appreciated."

---

> ### Author Response · Authors · 2023-11-22
>
> **Regarding the Novelty Concern**: We take the opportunity to stress that our primary innovation lies in the finding that, when the two-stage procedure is applied, the permutation-based objective of TACTiS [1] (Eqn (6)) is not needed to obtain valid attentional copulas, leading us to develop an objective with linear complexity (Eqn (7)). To detail the specific modifications to the underlying theory of the two-stage procedure, we remove all hypotheses about distributional assumptions and show that the two-stage procedure yields a valid solution to Problem (4): see Proposition 2 (proof in App. B.2). Further, we prove a new result showing the necessity of the two-stage approach (or an alternative method) for the non-parametric case: see Proposition 1. We will strive to make this very clear in the final revision and thank the reviewer for this feedback.
>
> **Regarding the Efficiency Contribution**: We agree with the reviewer that the efficiency improvement comes from the two-stage procedure.
> However, as for the improvements in the negative log likelihood, as shown in Figure 4, this is primarily attributed to the reduced parameter complexity (from factorial to linear) and the architectural innovation.
>
>
> [1] Alexandre Drouin, Etienne Marcotte, and Nicolas Chapados. TACTiS: Transformer-attentional copulas for time series. In Kamalika Chaudhuri, Stefanie Jegelka, Le Song, Csaba Szepesvari, Gang Niu, and Sivan Sabato (eds.), Proceedings of the 39th International Conference on Machine Learning, volume 162 of Proceedings of Machine Learning Research, pp. 5447–5493. PMLR, 17–23 Jul 2022.

---

> > ### Author Response · Authors · 2023-11-22
> >
> > **Regarding the difference in performance**:
> >
> > We are glad that you found our response satisfactory, and we thank you again for your detailed feedback. To answer your question, the joint distribution of TACTiS and TACTiS-2 can be broken down into two main modules:
> > * (1) components that model the marginal distributions and
> > * (2) components that model the copula distribution.
> >
> > TACTiS and TACTiS-2 differ significantly in (2). However, they are similar in terms of (1), as they both rely on Deep Sigmoidal Flows [1] for (1). Hence, it is reasonable to expect that their ability to model the marginal components of the distribution is similar. Regarding the slight difference in CRPS-Sum, we attribute it to the use of different sets of hyperparameters (tuned based on the negative log-likelihood), and to the fact that the CRPS-Sum captures some of the improvements in the multivariate dependencies, although being dominated by the marginals.
> >
> > We remain at your disposal for further clarifications or any other concerns to be addressed for you to consider revising your score.
> >
> > [1] Chin-Wei Huang, David Krueger, Alexandre Lacoste, and Aaron Courville. Neural autoregressive flows. In Jennifer Dy and Andreas Krause (eds.), Proceedings of the 35th International Conference on Machine Learning, volume 80 of Proceedings of Machine Learning Research, pp. 2078–2087. PMLR, 10–15 Jul 2018.

---

> > > ### Comment · Reviewer_NV8k · 2023-11-23
> > > **Response to the authors**
> > >
> > > Thank you for your response. If the hyperparameter difference is cited as a reason for CRPS metric improvement, does this not raise concerns about a fair evaluation? Shouldn't the baseline be optimized similarly for a more equitable comparison?

---

> > > > ### Comment · Reviewer_NV8k · 2023-11-23
> > > > **Overall review**
> > > >
> > > > I would like to thank the authors for their efforts in the response, which clarified many points for me. While I see the potential for the work, I still have concerns regarding overclaiming for novelty and efficiency part that is mainly attributed to the two-stage approach which is not new. More efforts need to be made to clarify their novelty over just adopting a two-stage approach to non-parametric settings. I would raise my score to 6.

---

> > ### Comment · Reviewer_NV8k · 2023-11-23
> > **Response to authors**
> >
> > Thanks to the authors for their response.
> > - Novelty: I recommend emphasizing your contribution separately from the two-stage approach, this would be more clear.
> >
> > - Efficiency: as you agree that efficiency improvement comes mainly from the two-stage approach, then you should not claim it is your contribution, as the two-stage procedure is not originally done by your side. I will suggest revising these claims on the paper, or mentioning that thanks to the two-stage approach, you have an efficiency improvement.

---

> ### Author Response · Authors · 2023-11-23
>
> > If the hyperparameter difference is cited as a reason for CRPS metric improvement, does this not raise concerns about a fair evaluation? Shouldn't the baseline be optimized similarly for a more equitable comparison?
>
> To clarify, we use exactly the same hyperparameter search protocol for both TACTiS and TACTiS-2 (see Appendix C.3). Of course, this does not guarantee that the optimal hyperparameters will be the same for both.
>
> > While I see the potential for the work, I still have concerns regarding overclaiming for novelty and efficiency part that is mainly attributed to the two-stage approach which is not new.
>
> Sec 3.2 in the paper has been revised to make it explicit that the two-stage approach has been previously studied in the copula literature (see text in blue above Eqn. 7). Of course, if the reviewer can point us to any specific lines in the paper which contain overclaims, we would be more than happy to rectify them.

---

### Author Response · Authors · 2023-11-17
**General Comment - Part 3**

### 3) Future Directions for the Community to Build On

Let us now discuss the broader significance of our work for the time series and machine learning communities. For the time series community, the flexibility and efficiency of TACTiS-2 make it a good starting point for a variety of extensions. First, one could explore variants based on large-scale transformers, which would extend it to support high-frequency measurements, longer forecasts, and a greater number of time series. Such extensions are facilitated by the linear (instead of factorial) complexity of our training objective. Second, TACTiS-2 could serve as a basis for foundation models for time series, a topic which has recently received considerable attention [6, 7, 8]. Being transformer-based, one could train an attentional copula on a wide variety of datasets, with the aim of inferring in a zero-shot fashion the dependencies between a new set of time series. The separation between marginal and copula components could further enable practitioners to train univariate probabilistic forecasting models and recombine them to produce multivariate forecasts.

As for the general machine learning community, we emphasize the applicability of non-parametric attentional copulas to **any** multivariate density estimation task, not only for time series. It is for this reason that in Section 3 we present non-parametric attentional copulas as a distinct topic of interest. We believe that our new, two-stage, approach to learning non-parametric attentional copulas will unlock their application to a wider variety of high-dimensional density estimation problems (e.g., natural language processing, biological data).

We kindly invite the reviewers to re-assess the paper in light of these clarifications.

## References

[1] Alexandre Drouin, Etienne Marcotte, and Nicolas Chapados. TACTiS: Transformer-attentional copulas for time series. In Kamalika Chaudhuri, Stefanie Jegelka, Le Song, Csaba Szepesvari, Gang Niu, and Sivan Sabato (eds.), Proceedings of the 39th International Conference on Machine Learning, volume 162 of Proceedings of Machine Learning Research, pp. 5447–5493. PMLR, 17–23 Jul 2022.

[2]  Marin Bilos, Kashif Rasul, Anderson Schneider, Yuriy Nevmyvaka, and Stephan Gunnemann. Modeling temporal data as continuous functions with stochastic process diffusion. In Andreas Krause, Emma Brunskill, Kyunghyun Cho, Barbara Engelhardt, Sivan Sabato, and Jonathan Scarlett (eds.), Proceedings of the 40th International Conference on Machine Learning, volume 202 of Proceedings of Machine Learning Research, pp. 2452–2470. PMLR, 23–29 Jul 2023.

[3] Joe, Harry, and James Jianmeng Xu. 1996. “The Estimation Method of Inference Functions for Margins for Multivariate Models.” R. Faculty Research and Publications. October 31.

[4] Andersen, Elisabeth Wreford. “Two-stage estimation in copula models used in family studies.” Lifetime Data Analysis 11 (2005)

[5] Joe, Harry. “Asymptotic efficiency of the two-stage estimation method for copula-based models.” Journal of Multivariate Analysis 94.2 (2005).

[6] Rasul, K., Ashok, A., Williams, A. R., Khorasani, A., Adamopoulos, G., Bhagwatkar, R., ... & Rish, I. (2023). Lag-Llama: Towards Foundation Models for Time Series Forecasting. arXiv preprint arXiv:2310.08278.

[7] Das, A., Kong, W., Sen, R., & Zhou, Y. (2023). A decoder-only foundation model for time-series forecasting. arXiv preprint arXiv:2310.10688.

[8] Chin-Chia Michael Yeh, Xin Dai, Huiyuan Chen, Yan Zheng, Yujie Fan, Audrey Der, Vivian Lai, Zhongfang Zhuang, Junpeng Wang, Liang Wang, and Wei Zhang. 2023. Toward a Foundation Model for Time Series Data. In Proceedings of the 32nd ACM International Conference on Information and Knowledge Management (CIKM ‘23). Association for Computing Machinery, New York, NY, USA, 4400–4404.

---

### Author Response · Authors · 2023-11-17
**General Comment - Part 2**

## Significance of Contribution

Reviewers NV8k and s9Tw have raised concerns about the significance of this work, questioning its value for the broader time series community. This perception likely arises from the fact that our work does not introduce new modeling components (e.g., attentional copulas) or new capabilities (e.g., greater flexibility). While these comments are well-founded, we now argue for the significance of our work based on three aspects: 1) it creates tangible value for the time series community, through 2) non-trivial contributions, and 3) presents results on which the community will be able to build.

### 1) Tangible Value for the Time Series Community

**Accuracy**: As discussed in the introduction, “general-purpose models” that are suitable for a variety of tasks and applications are of high value for the broad time series community since they remove from practitioners the burden of making modeling choices. In the related works, we mention that there currently exist only two models that qualify as such: TACTiS [1] and SPD [2]. The results reported in Table 1, 4, and 5 clearly demonstrate the superior predictive performance of TACTiS-2 over SPD. Further, these results, combined with Tables 2-3 and the additional results reported in the revised paper in Tables 12-14, clearly demonstrate the superiority of TACTiS-2 over TACTiS, a fact that was recognized by all reviewers. This establishes our proposed TACTiS-2 model as the most accurate general-purpose model available to practitioners.

**Efficiency**: Further, the empirical results reported in Figure 4 and App. A.3 show that TACTiS-2 converges to better solutions than TACTiS using much fewer FLOPs. This improvement is significant given that TACTiS’ training run had a duration of **three days**. We realize that this fact was not clear in the initial submission and have emphasized it in the revised paper.

Hence, these results establish TACTiS-2 as the most accurate general-purpose method available to the community. We believe that this, combined with its superior efficiency, make TACTiS-2 cross an inflection point in applicability, bringing the desirable theoretical properties of TACTiS into the realm of tangible real-world impact.


### 2) New Knowledge and Non-Trivial Contributions

TACTiS-2 results from a series of non-trivial methodological contributions:
1. Two-stage optimization of non-parametric copulas: We prove that valid non-parametric copulas can be learned without the costly permutation-based objective of Drouin et al. (2022) [1], reducing the number of parameters from O(d!) to O(d). We agree with the reviewers that the two-stage approach has been explored in the copula literature. In fact, we duly attribute it to Joe and Xu (1996) [3] in Section 3.2. However, to the best of our knowledge previous work has only shown the correctness of this approach for the case of parametric [3, 4, 5] (where the marginals and copula use parametric forms) and semi-parametric [4] (where the marginals use a non-parametric form and the copula uses a parametric form) settings. In Proposition 2 (proof in App B.2), we show the validity of this approach for the non-parametric case that is of interest to our work and to future works on non-parametric density estimation. Further, we propose a new theoretical result that motivates the need for such a training approach in the context of non-parametric copula-based density estimation (Proposition 1, proof in App B.1).

2. Practical implementation: While our model builds on TACTiS, we radically change the optimization problem and training procedure, proposing a training curriculum that demands subsequent restructuring to the architecture (e.g., the introduction of a dual encoder). These changes could not be trivially deduced from the work of Drouin et al. (2022), and they entail substantial improvements in prediction accuracy and efficiency.

---

### Author Response · Authors · 2023-11-17
**General Comment - Part 1**

# General comment

We are grateful to the reviewers for their thoughtful feedback and careful evaluation of our paper. We are encouraged by the overall positive tone of the reviews, which recognize the soundness of our work and the clarity of the exposition.

The reviewers have noted that our work addresses “a challenging problem […] which has broad applicability across numerous fields” (NV8k). Further, they recognize that our work addresses a “key limitation” (s9Tw) of previous work and that the “proposed technique is well-motivated” (sGoi). Finally, reviewers NV8k, sGoi, and GAmj acknowledge the state-of-the-art forecasting performance of TACTiS-2, and all reviewers observe that TACTiS-2 shows clear gains in predictive performance and training dynamics over the state-of-the-art TACTiS model.

We appreciate the diverse critical feedback we received and have revised the paper accordingly. In what follows, we start by addressing concerns regarding the significance of this work. We then address the concerns of each reviewer separately and provide additional clarifications and results where needed. We hope that all reviewers will find this response satisfactory and look forward to engaging in deeper discussions.

---

### Meta-Review · Area_Chair_aY58 · 2023-12-10

**Metareview:**

This paper presents TACTiS-2, an improved version of TACTiS, which has a revised architecture trained by the two-stage solution. Before the rebuttal, the reviewers raised various concerns, including distinctiveness from prior work, the clarity of the proposed method, significance beyond incremental improvements, and the depth of analysis. I agree with the reviewers about these concerns, however, most of these concerns have been addressed by the authors during the rebuttal. Therefore, I am inclined to accept this paper.

**Justification For Why Not Higher Score:**

There are still some issues from the reviewers after the rebuttal. Myself also think the evaluation can be further improved (e.g., more recent baselines can be included). That is why we can not recommend a higher score.

**Justification For Why Not Lower Score:**

The majority of the reviewers are positive on this paper. Other two reviewers giving scores of 5 mentioned that the paper has been improved after the rebuttal. Considering that most of the concerns have been addressed during the rebuttal, I would like to recommend accepting this paper.

---

### Decision · Program_Chairs · 2024-01-16

Accept (poster)